# Ports' criticality in international trade and global supply-chains

J. Verschuur [1] ✉, E. E. Koks [1,2] & J. W. Hall [1]

We quantify the criticality of the world's 1300 most important ports for global supply chains by predicting the allocation of trade flows on the global maritime transport network, which we link to a global supply-chain database to evaluate the importance of ports for the economy. We find that 50% of global trade in value terms is maritime, with low-income countries and small islands being 1.5 and 2.0 times more reliant on their ports compared to the global average. The five largest ports globally handle goods that embody >1.4% of global output, while 40 ports add >10% of domestic output of the economies they serve, predominantly small islands. We identify critical cross-border infrastructure dependencies for some landlocked and island countries that rely on specific ports outside their jurisdiction. Our results pave the way for developing new strategies to enhance the resilience and sustainability of port infrastructure and maritime trade.

Maritime transport is considered the backbone of international trade and the global economy[1–3]. With ports supporting the integration of production centres and consumer markets across borders[4], there are large dependencies and feedbacks between changes in the size and structure of the economy (e.g. trade composition, supply-chain structure) and the expected freight flows through specific ports[5,6]. Similarly, changes (e.g. new infrastructure investments) or disruptions (e.g. port closures) to the maritime transport network can have implications for supply-chains across multiple countries and industries[7].

The maritime transport and global supply-chain networks interact with one another on different spatial scales, with recent events illustrating the tight coupling between the two. On the largest spatial scale, the global trade network, the demand for maritime trade is driven by countries' demand for trade, those countries supplying this trade, and the share of trade being maritime (i.e. modal split). Hence, relative changes in freight flows reflect changes in trade demand, supply and modal split. The COVID-19 pandemic, which affected port operations across the world, changed both demand and supply patterns simultanously[8]. On the one hand, this disrupted maritime transport and supply-chains due to factory shutdowns, port closures and labour shortages[9], while on the other hand this led to large trade bottlenecks at many ports due to shifting demand patterns[10]. Freight demand on

the underlying maritime transport network, consisting of maritime routes that connect ports, is determined by the geographical demand for transport services and the network structure of system. For certain commodities this network is known to be more centralised (e.g. containers) while for others this is more decentralised (e.g. bulk transport)[11]. The 2021 Suez blockage highlighted how a large shock to a specific route within the maritime transport network could affect multiple ports across the globe, and eventually supply-chains depending on these ports[12]. Ultimately, trade flows handled at the port serve supply-chains across different hinterlands, either directly (e.g. firm directly receiving goods from ports) or indirectly (e.g. firm depending on other firms that receive goods from ports). For instance, Hurricane Katrina (2005), shutting down major Louisiana ports, led to large disruptions to the global grain supply, resulting in export losses for the United States, which rippled to dependent supply-chains globally and raised commodity prices[13,14].

The criticality, that is the systemic (i.e. network-based) importance, of ports for the economy is often framed in terms of the absolute amount of trade flowing through a port, its network characteristics within the maritime transport network (e.g. node centrality)[11,15,16], or in terms of its contribution to the local or regional economy (e.g. regional employment and value-added)[17]. These framings, however, ignore the primary function of ports as the physical

[1]Environmental Change Institute, University of Oxford, Oxford, UK. [2]Institute for Environmental Studies, Vrije Universiteit Amsterdam, Amsterdam, Netherlands. ✉e-mail: jasper.verschuur@keble.ox.ac.uk

infrastructure that connects supply-chains across countries[1,4], and therefore fail to provide a comprehensive picture of the dependencies and feedbacks between ports and the economy.

Establishing a fine-scale representation of how each individual industrial sector, globally, makes use of maritime transport, and, on the other hand, how individual ports are critical to global supply chains can help us rethink the importance of ports, which can be informative for different disciplines. For instance, it could allow a better understanding of the geographical distribution of physical trade flows across supply-chains[18,19], connect environmental footprints with commodity flows[20,21], predict future port demand (in terms of volume and space required) as economies grow[22], help allocate maritime emissions (~2.6% global greenhouse gas emissions in 2012) to countries and sectors[23,24], and assess the potential supply-chain losses due to maritime transport disruptions[25,26].

So far, a number of macroeconomic studies have examined the evolution of international trade and supply-chain interconnectivity[27–29]. This analysis is backed by advances in the provision of Multi-Regional Input–Output (MRIO) tables that describe the inter-, and intra-industry dependencies within countries and between countries[30–33]. Although MRIO tables provide extensive data on inter-, and intra-industry trade flows, at national and regional scales, it does not provide insights into the domestic and international transportation systems that are used for these trade flows. Another strand of literature has analysed the network structure and evolution of maritime transport networks through a complexity science lens[11,15,34–39]. This research, however, focused solely on the shipping connections between ports, without incorporating information on the goods that are carried by maritime vessels, where goods are coming from and going to, and how goods are used in the economy. Hence, to date, there is still a spatial mismatch between information describing the structure of the global economy (i.e. global

trade and supply-chain data) and a bottom up representation of the transportation network used (i.e. observed maritime transport flows) to facilitate this economic structure.

Here, we present a new modelling framework that provides a comprehensive understanding of the different dimensions of the criticality of ports for domestic and global economies (e.g. on the trade, transport and supply-chain level) that are not captured in aggregate port-level trade statistics. To do this, we provide a globally consistent assessment of the links between ports and maritime trade, the transport networks they utilise (1378 ports across 207 countries), and the supply-chains they serve (1298 ports across 176 countries) (see Methods). This is achieved by first estimating the fraction of maritime trade in all bilateral trade flows and feeding this into the newly developed Oxford Maritime Transport (OxMarTrans) model that simulates the maritime and hinterland routes taken to transport maritime trade flows. The trade flows going through ports are then linked to a global supply-chain database (EORA MRIO tables[32]) to quantify the links and feedbacks between ports and the economy.

We find that around 50% of global trade (in value terms) is maritime, which reaches up to 76% for the mining and quarrying sector. Low income countries and small island developing states (SIDS) rely disproportionally on maritime trade: their maritime import fraction is 1.5 and 2.0 times higher, respectively, than the global average. Every USD flowing through a port contributes on average 4.3 USD in value to the global economy. We identify ports being critical for the global and domestic economy, showing how the top 5 macro-critical ports all handle goods that contribute >1.4% to the global economy, while 40 ports handle goods that represent >10% of the value of the domestic economies they serve (i.e. domestically critical ports). In addition, we find that every 1000 USD increase in final demand (i.e. the goods needed to meet final consumption and exports) results in a median

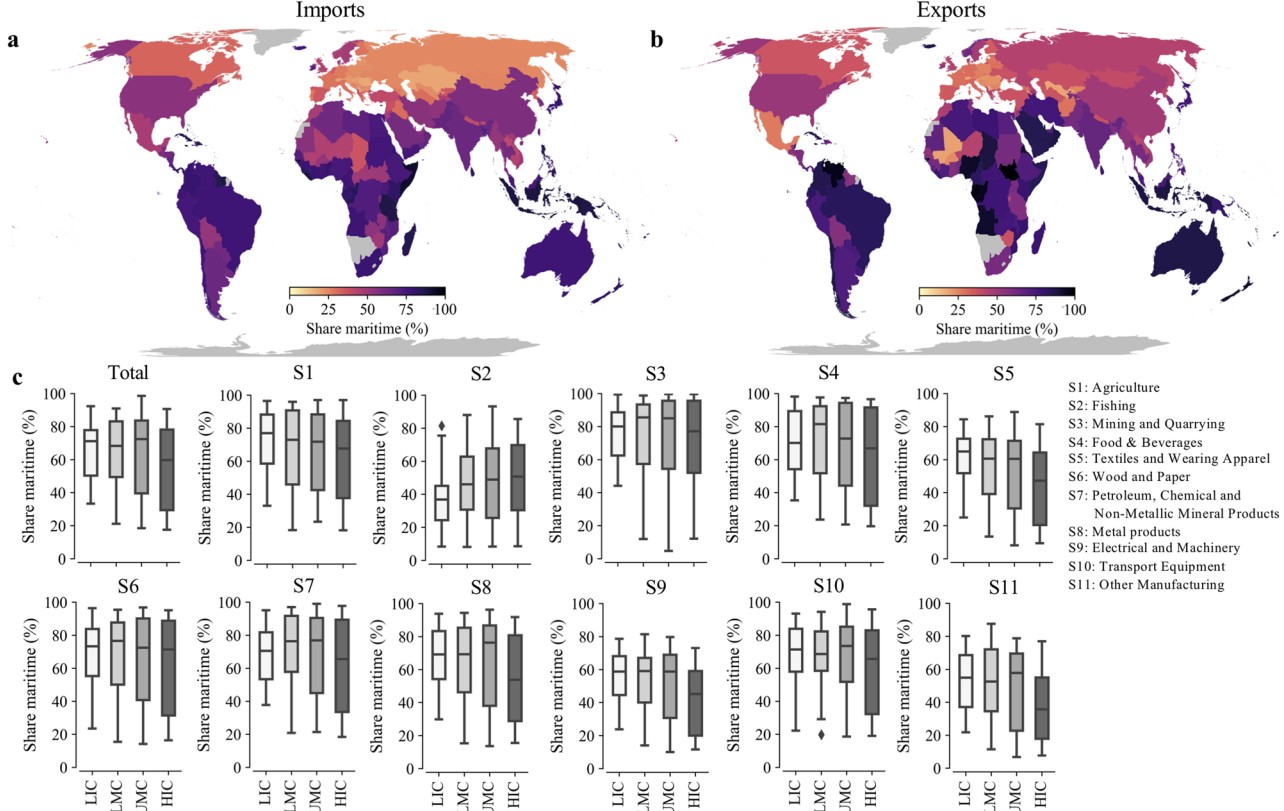

**Fig. 1 | The share of maritime transport in global trade.** Country's percentage of maritime imports (**a**) and exports (**b**) based on the 2015 trade network. **c** Boxplots of the percentage maritime imports per economic sector with countries grouped

by income level (based on the World Bank income classification). LIC: Low income countries, LMC: Lower middle income countries, UMC: Upper middle income countries, HIC: High income countries.

84.6 USD increase in maritime imports across the ports that serve these economies, with 30 individual ports experiencing >100 USD increase. Our results pave the way for a better understanding of the key links, dependencies and feedbacks between port, the maritime infrastructure network and the global economy, which is essential information for sustainable infrastructure planning.

## Results

### Overview
The results summarise the model output on the different layers; the trade network layer, the transport network layer, and the port supply-chain layer. These three layers are conceptualised in Supplementary Fig. 1. The trade network layer results discuss the output of the global modal split model (i.e. the distribution of trade flows across transport modes) that quantifies the variations in a country's dependency on maritime trade as a fraction of total trade on a commodity level. The transport network layer results outline several output of the OxMarTrans model. The OxMarTrans model simulates the route choice of millions of maritime freight flows between 3400 regions across 207 countries on the hinterland and maritime transport network. The output includes the aggregate global freight flows on the transport network and through the two main canals (Suez and Panama), the dependency of countries on maritime infrastructure in foreign jurisdictions through land-based connections and transhipments, the port-level trade flows, and the trade flow distribution across all ports. To quantify the domestic and global economic dependencies on trade flows through ports (i.e. the port supply-chain layer), we use the EORA MRIO tables[32] that we extend to the port-level to link the commodities that flow through ports the global supply-chains they serve. Two metrics are constructed to capture these dependencies; (1) the port-level output coefficient (PLOC) and (2) the port-level import coefficient (PLIC). The base year considered in this analysis is 2015, which is the latest available year in the EORA MRIO database (at the time of writing). Throughout this study, we adopt a 11 sector industry classification in line with the EORA MRIO to evaluate differences in criticality between sectors (Supplementary Table 1).

### Share of maritime transport in global trade
Within the trade network layer, the amount of maritime trade between countries is determined by the absolute value of trade across all modes between country pairs and the share of this being maritime. Our transport modal split model estimates the share of maritime trade for around 8 million bilateral trade flows globally on a commodity level (HS6). It should be noted that in this study the mode of transport is defined as the dominant transport mode (longest distance) in the supplier-consumer connection, which means that landlocked countries can still rely on maritime transport (see Methods).

We estimate that 9.4 billion tonnes of trade, equivalent to around 7.6 USD trillion in value terms, was maritime in 2015. The share of maritime trade in global trade is around 75% in terms of weight and 50% in terms of value. This number corresponds well with the estimated 9.96 billion tonnes of trade being discharged in ports in 2015 as reported by UNCTAD[40]. However, large differences exist between sectors. For instance, while 75.7% (86.0%) of Mining and Quarrying (sector 3) products are transported by means of maritime transport in value (weight) terms, most manufacturing sectors (sector 4 to 11) transport only 40% – 57% (53% – 60%) of their trade in value (weight) terms using maritime transport.

Figure 1 shows the percentage of maritime transport in total imports (Fig. 1a) and total exports (Fig. 1b) per country, while Supplementary Figs. 2 and 3 display the same results per economic sector considered. The dominance or absence of maritime transport for trade is mainly determined by the geographical location of trading partners (e.g. distance, island state), the presence of alternative (fast and cheaper) modes, the value to weight ratio of the commodities, and the

standard of living of the importing country (e.g. quality of logistics services)[41].

As can be seen from Fig.1, Caribbean islands, countries in Oceania and some countries in Africa (e.g. Somalia, Nigeria, Gabon) rely disproportionally on maritime transport for both imports and exports (Fig. 1a, b). European countries, in particular landlocked countries (e.g. Romania, Hungary, Switzerland), have a much lower share of maritime transport, mainly due to the large trade flows between European countries that use road, rail and inland waterway transport to move goods over relatively short distances[42,43]. Middle-Eastern (Saudi Arabia, United Arab Emirates) and South American (e.g. Brazil, Colombia) countries rely more on maritime transport for their exports compared to their imports. These countries mainly export raw materials (e.g. oil, coal, grain) which is predominantly shipped by maritime vessels, but import a more diversified mix of goods that are transported by multiple modes. Small Island Developing States (SIDS) rely disproportionally on maritime transport, with 86.5% of imports and 79.8% of exports being maritime, thus almost twice as much as non-SIDS countries. SIDS are often served by a only a few maritime transport routes and experience high transportation costs[44], making reliable maritime transport services critical for the well-functioning of SIDS' economies.

Figure 1c shows the share of maritime transport in total and sector-specific imports grouped by the income level of countries (using the 2021 World Bank classification). Low income countries import on average 1.5 times more by means of maritime transport compared to high-income countries (68% versus 45%). The difference is largest for the manufacturing sectors (sector 8 to 11), having maritime shares 1.5 – 1.8 times higher than high income countries. This difference can be explained by the fact that low income countries often trade low value bulk goods, for which maritime transport is the only viable option, and relatively few high valued goods that are more often transported by aeroplane[45]. Even within the same continent, such as in Africa, maritime transport is often the only feasible mode of transport for certain goods as the road infrastructure lacks the reliability and capacity for efficient trucking, and border crossings can be time consuming[46,47]. Therefore, the integration of low income countries into complex manufacturing supply-chains, which critically depend on just-in-time logistics services[48], could be hindered by their overreliance on maritime transport, which is considerably slower than air transport[49,50].

### Global maritime transport flow allocation
The maritime transport network, consisting of ports and maritime routes transporting goods using different vessel types (e.g. tankers, containers), connects the locations of production to their demand markets. The OxMarTrans model predicts which ports and maritime routes, including locations of transhipments, are being used to transport the maritime trade flows between each country pair and per economic sector (see Methods). The underlying hinterland and maritime consists of 1378 ports, with the port connections and maritime network capacities incorporated in the model based on a dataset of observed ship activities from Automatic Identification System (AIS) data[9]. The OxMarTrans model therefore helps identify the spatial connectivity of ports; the maritime subnetwork that is used to transport goods from and to a specific port (we show the spatial connectivity for nine ports in Supplementary Fig. 4).

Globally, to meet maritime trade demand, we estimate that 90.5 trillion tonnes-km of freight is transported across sea and an additional 33.4 trillion tonnes-km over land to connect hinterlands to ports. The maritime freight predicted by the model consistent with the 84 trillion tonnes-km estimated by UNCTAD[40]. 43% of the total maritime tonnes-km is attributed to the Mining and Quarrying sector (sector 3) alone, while the manufacturing of Electrical and Machinery products (sector 9), Transport Equipment (sector 10) and Other Manufacturing goods (sector 11) together account for only 2.7% of total tonnes-km.

Supplementary Fig. 5 shows the total throughput (sum of import, export and transhipment) per port and estimated flows on the maritime transport network, while Supplementary Fig. 6 shows a similar result per sector.

Many countries depend on the transport of goods passing the Suez or Panama canal. In total, our model predicts that around 1.1 USD trillion (13.9% of maritime trade) and 0.49 USD trillion (6.2% of maritime trade) pass the Suez and Panama canal, respectively, in 2015 in

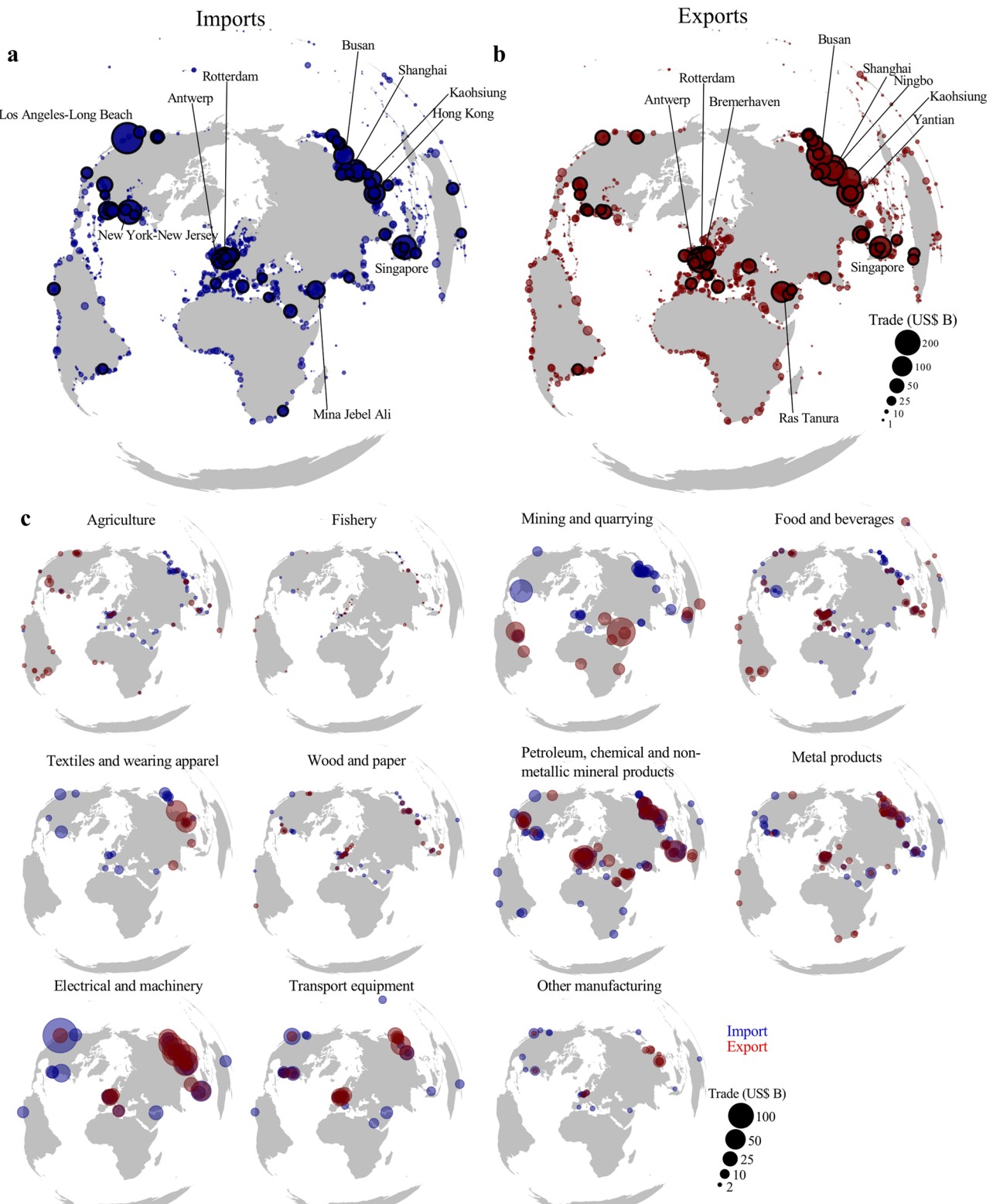

**Fig. 2 | Cross-border maritime infrastructure dependencies. a** The port-level throughput (import, exports and transhipment) that comes from or goes to ports by passing the Panama canal. **b** Same as (**a**) but for the Suez canal. **c** The share and absolute value of port-level throughput that is linked to a foreign economy, either because of transhipment or a land-based connection. Some regionally important ports in terms of foreign dependencies are annotated.

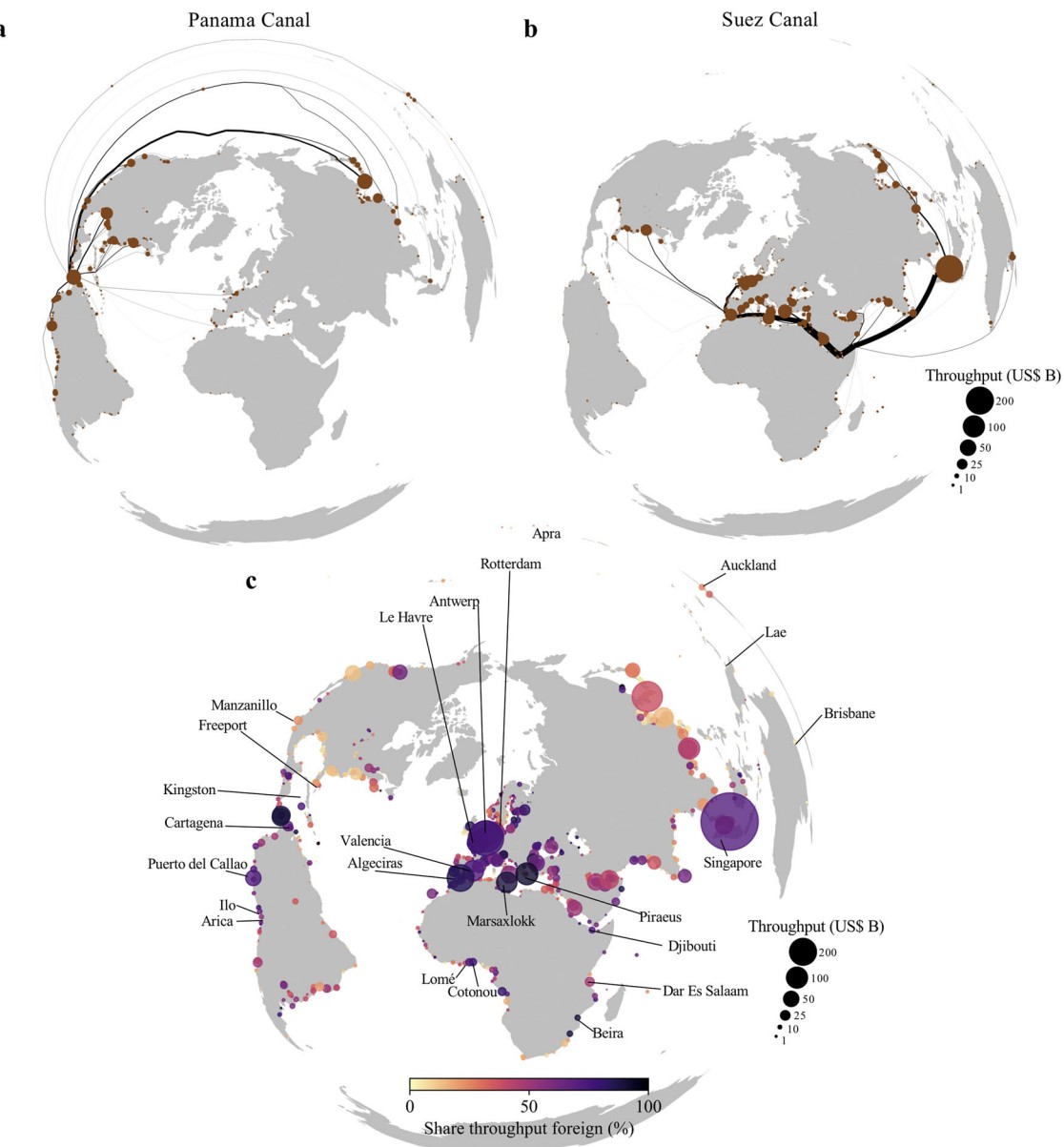

**Fig. 3 | The origin and destination ports of trade flows.** The aggregated imports (**a**) and exports (**b**) per port. The critical ports are highlighted with the top 10 ports annotated. **c** The location of the critical importing (blue) and exporting (red) ports per sector.

line with official statistics (see Supplementary Note 3). For the Panama canal, ports in the Gulf of Mexico, the west coast of South America, and parts of East Asia rely directly on goods being shipped through the canal (Fig. 2a). The Suez canal is important for trade going from and to the Asia and Europe. On the east of the canal, the ports of Singapore, Jeddah, Colombo, Mina Jebel Ali are most dependent on the Suez canal, while on the west of the Suez canal the ports of Piraeus, Rotterdam, Marsaxlokk and Algeciras rely most on it (Fig. 2b).

**Cross-border maritime infrastructure dependencies**

Both landlocked and maritime economies rely on maritime infrastructure in other countries because they either use ports in neighbouring countries to import or export goods, or they use transhipment services to ship goods from origin to destination. For instance, around 28% of the world's container throughput in 2012 involved transhipment, where containers unloaded from a deepsea vessel are being transhipped to another deepsea vessel or a smaller vessel (i.e. feeder vessels) to serve otherwise unconnected port pairs[51].

Using the OxMarTrans, we estimate that approximately 16.4% of global port throughput (in value terms) is transhipped, while 19.4% of port throughput are imports to or exports from foreign countries connected via the hinterland transport network. Figure 2c shows the fraction of port throughput being foreign per port. In absolute terms, large transhipment hubs (Singapore, Algeciras, Valencia and Marsaxlokk) have a high share of foreign throughput. Additionally, ports in the Le Havre-Hamburg range (Le Havre, Antwerp, Rotterdam, Bremen) handle the largest amount of foreign import and export value, as they compete for trade going to, and coming from, the Central European hinterland[52].

Regionally, some ports play key roles in serving landlocked countries or island states (see highlighted port in Fig. 2c). In Africa, for instance, the port of Djibouti handles almost all of Ethiopia's maritime trade, the ports of Dar Es Salaam (Tanzania) and Beira (Mozambique) are essential for landlocked countries in Sub-Saharan Africa, while the port of Lomé (Togo) and Cotonou (Benin) are key for Western-African landlocked countries. In South America, the ports of Arica (Chile) and

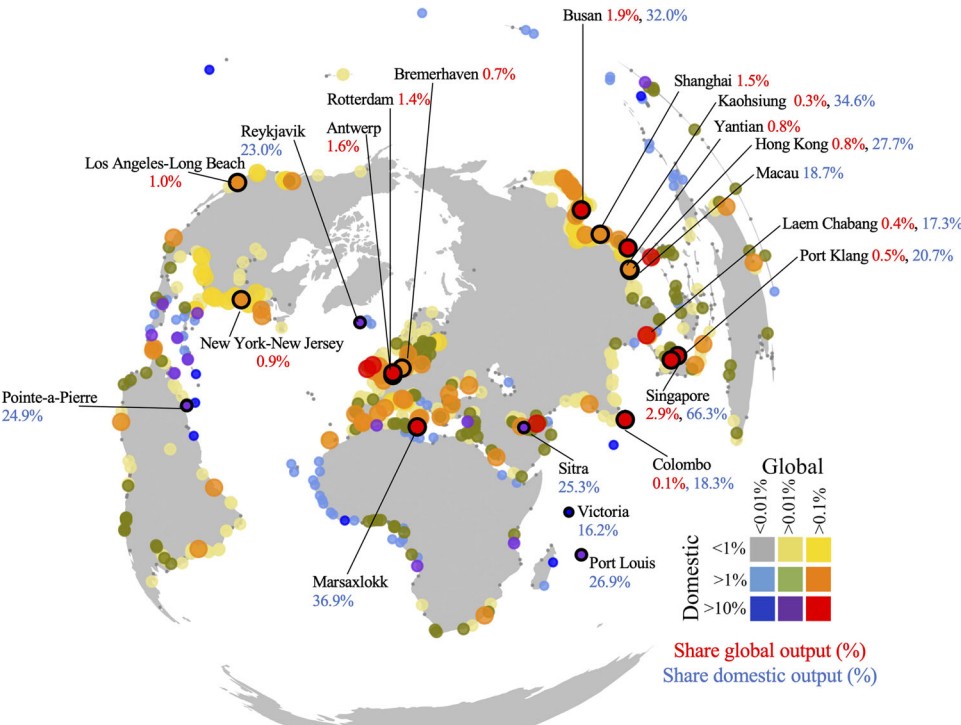

**Fig. 4 | Distribution of the domestically and globally critical ports in terms of industry output.** The importance of trade flows going through ports in terms of its contribution to the domestic output as a percentage of total domestic output and global output as a percentage of total global output. The ten ports most critical ports in terms of domestic and global output are highlighted together with the associated percentage value (domestic in blue, global in red).

Ilo (Peru) handle the majority of maritime trade of Bolivia, while Puerto del Callao (Peru) is an important transhipment hubs for South America. In Oceania, several ports (e.g. Brisbane, Auckland, Apra, Lae) serve as important transhipment hubs for Pacific island economies, with a similar observation for key regional transhipment hubs in the Caribbean region (see Fig. 2c).

## Distribution of trade flows per port

Several factors determine the total maritime trade flows going through ports (e.g. maritime connectivity, logistics services, presence of hinterlands). Figure 3a, b shows the distribution of imports (Fig. 3a) and exports (Fig. 3b) across all trade flows, with the top 10 largest ports annotated. We also show the global core ports, defined as those ports responsible for importing or exporting 50% of global trade (black edge colour). Core importing ports are located in North-America (Los Angeles-Long Beach, New York-New Jersey), Western Europe (Rotterdam), the Middle-East (Mina Jebel Ali) and Asia (Singapore, Shanghai) that serve the populated hinterlands (so-called gateway ports[39]) or industrial and logistics hubs. Among the core exporting ports are specialised ports that are critical for the exports of agricultural products (Vancouver, New Orleans, Santos), petrochemicals (Houston, Singapore, Rotterdam), iron ore (Port Hedland and Dampier), electrical and machinery manufacturing (Shanghai, Busan, Kaohsiung), car manufacturing (Ulsan, Nagoya, Bremerhaven), and oil and gas (Ras Tanura, King Fahad Industrial Port).

Trade is highly concentrated in a relatively small number of core ports. The trade unevenness expresses the number of ports that handle 10%, 50% and 90% of trade. Only 4 (3) ports are responsible for 10%, 56 (48) ports are responsible for 50%, while 378 (366) ports are accounting for 90% of global maritime imports (exports) (Supplementary Table 2). This underlines that from a global perspective, the maritime transport network consists of a small number of core ports and a large number of secondary (i.e. periphery) ports.

The aggregate results do hide the importance of certain ports on a sector level. Figure 3c shows the geographical location of the core importing and exporting ports per sector, showing a clear geographical clustering of trade flows that are either connected to important demand markets[53], or closely located to large sector-specific industry clusters[53]. Agriculture trade (sector 1) has clear origin ports in the United States, Brazil and Argentina, serving ports in Europe and across Asia. The import and export hotspots of Mining and Quarrying (sector 3) and Food and Beverages (sector 4) products are more spread across the globe, reflecting the export specialisation of different regions (e.g. oil in Middle-East, iron ore and coal in Australia, food products in Indonesia and Malaysia). The Wood and Paper manufacturing (sector 6) sector has large exporting ports in Scandinavia, the United States and China, that export timber products to ports in the United Kingdom, Japan and the Middle-East. Metal products (sector 8) are exported through Chinese, South African and Chilean ports and supplied to the Middle-East, South-East Asia and the United States. The remaining manufacturing sectors (sector 5, 9-11) all have large exports in ports in Western-Europe, East-Asia and the United States, with goods imported in ports in the Middle-East, Australia and parts of South America.

The trade unevenness differs considerably per sector (see Supplementary Table 2). The largest unevenness is found for the exports of Textiles and Wearing Apparel (sector 5), manufacturing of Transport Equipment (sector 10) and Other Manufacturing (sector 11) while the lowest level of trade unevenness is found for the imports of Agricultural products (sector 1), Food and Beverages (sector 4), and Petroleum, Chemical and Non-Metallic Mineral products.

These sectoral heterogeneities do not only reflect the differences in the clustering of industries, but also economies of scale present in the transport of some goods[54,55]. For example, while for some highly concentrated sectors the vast majority of goods will be transported between a subset of core ports, other less concentrated sectors will use

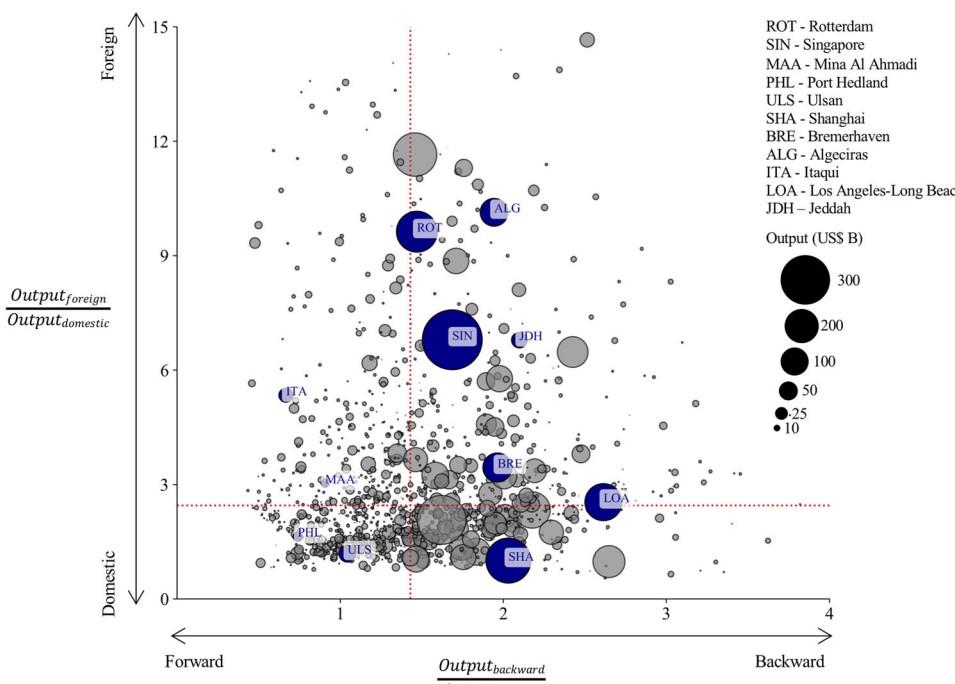

**Fig. 5 | The relative importance of forward/backward and domestic/foreign port-industry linkages.** The contribution of port-level trade to total output subdivided into forward and backward economic linkages and domestic and foreign economic linkages, capturing the relative importance of the four components. The size of the dot corresponds to the total output linked. The red dotted line depicts the median values across all ports. Ports highlighted in blue and annotated are mentioned in the text.

a more decentralised transport network. These sectoral differences reinforce the results found in previous studies that analysed the characteristics of networks of different types of maritime vessels (which are indicative of the sector) and found similarly critical differences between these vessel networks[11,35,39].

## Port-level output coefficient

Every port is connected to one or multiple supply-chains in the domestic and foreign economies they serve, either through direct (e.g. through firms directly sending or receiving goods from a port) or indirect (e.g. through firms depending on other firms that send or receive goods from a port) economic connections. More specifically, the products that are imported through a port are either directly consumed in an economy or are used in production processes to produce goods for domestic consumption or export. Additionally, goods exported through a port are being used in production processes, or directly consumed, elsewhere. We call this the port supply chain network. To understand the criticality of the trade facilitation function of ports for domestic and global supply-chains, we developed a metric, called the port-level output coefficient (PLOC), that captures the total industry output and consumption directly or indirectly dependent on the trade flows through a port, either in absolute terms (PLOCA) or relative to the amount of trade going through a port (PLOCR). This is done by removing the trade flows going through a port from the extended MRIO table and quantifying the output changes to the domestic and global economy (see Methods).

In relative terms (PLOCR), every USD of trade going through a port influences on average (5th–95th percentiles) 4.34 (3.84 – 5.03) USD of value in the global economy (Supplementary Fig. 7). Large relative values are found for ports in East-Asia (e.g. China, South-Korea, Taiwan), which are strongly integrated in global supply-chains, but also for some of the raw materials exporting ports in Australia (e.g. Port Hedland and Dampier) and Africa (e.g. Port of Saldanha), which are important for supply-chains downstream (e.g. firms using intermediate products that are produced using raw materials).

In absolute terms (PLOCA), some ports are important for the domestic economy, while others are more important for the global economy. In some cases, ports are critical for both, as outlined in Fig. 4a, which shows the top 10 most critical ports for the domestic economy and the global economy. The top 5 most critical ports for the global economy (Singapore, Shanghai, Busan, Rotterdam, Antwerp) all handle goods that directly or indirectly contribute to >1.4% of global industry output. In total, 94 ports are considered macro critical for global supply-chains, indicating that more than 0.1% of global industry output depends on these ports. 40 ports are considered domestically critical, with over 10% of industry output dependent on trade going through a single port. Examples of some ports that are critical for the domestic economy but negligible on a global scale (dark blue or purple markers Fig. 4a) are the ports of Port Louis (Mauritius, 26.9% of domestic output), Pointe-a-Pierre (Trinidad and Tobago, 24.9% of domestic output), Reykjavik (Iceland, 23.0% domestic output) and Sitra (Bahrain, 25.3% of domestic output). The ports of Kaohsiung (Taiwan), Hong Kong (Hong Kong), Laem Chabang (Thailand), and Port Klang (Malaysia) (red markers Fig. 4a) are found to be essential for both the domestic and global economy. A similar figure can be produced for the final consumption needs of countries, with globally and domestically critical ports shown in Supplementary Fig. 8. Although an overall similar spatial footprint, some ports are more important for meeting final consumption, especially for some small island economies where single ports import over 35% of the final consumption requirement. Hence, the tendency to focus on the absolute size of trade going through a port to classify its importance ignores how some smaller ports are still critical for domestic economies.

## Position port in global supply-chains

To unpack the PLOC metric even more, one can characterise whether the goods that flow through a port are relatively more dependent on domestic or foreign production processes, and relatively more on forward (exporting goods being used in production processes downstream in the supply-chain) or backward linkages (import goods that

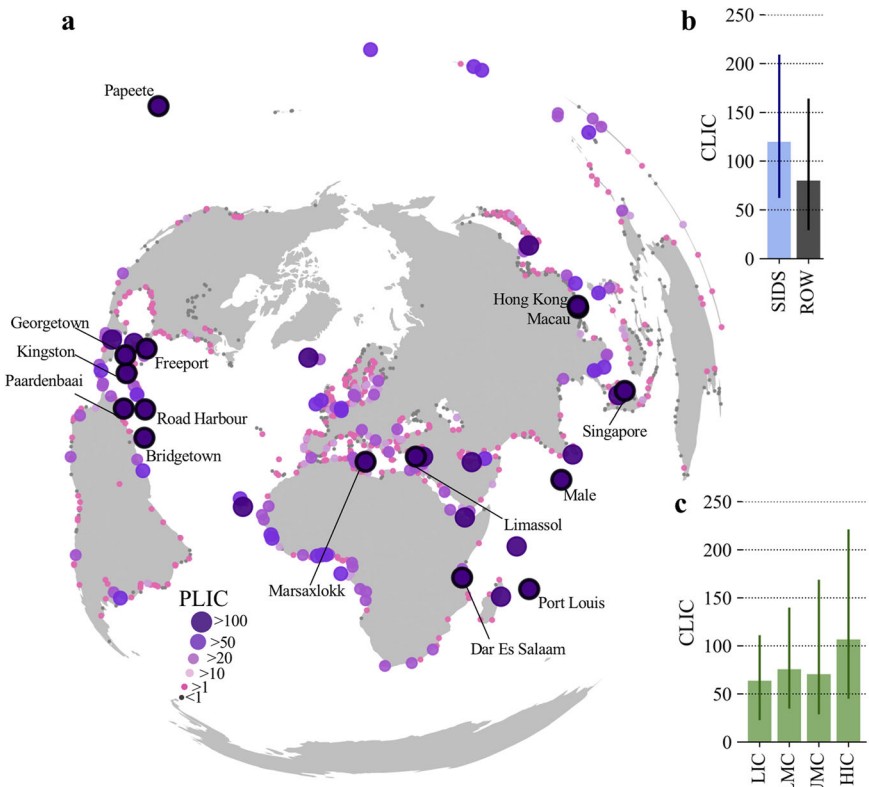

**Fig. 6 | Global distribution of the country-level and port-level import coefficient. a** The global distribution of the port-level import coefficient (PLIC), expressing the USD increase in imports for every 1000 USD increase in final demand. The top 15 ports are highlighted and annotated. **b** The country-wide maritime import coefficient (CLIC) for Small Island Developing States (SIDS) and to the rest of the world. **c** Same as (**b**) but the CLIC of the countries grouped by income level (based on the World Bank income classification). LIC: Low income countries, LMC: Lower middle income countries, UMC: Upper middle income countries, HIC: High income countries.

are produced using production processes upstream in the supply-chain). The relatively importance of these four components determine how ports are positioned differently within the global supply-chain network.

In Fig. 5, we show the relative importance of port throughput in terms its contribution to industry output downstream (forward) or upstream (backward) in the supply-chain and the degree to which output is linked to domestic or foreign supply-chains. We show the position of a number of ports that are all considered macro-critical but located at opposite ends of the spectrum. The ports of Rotterdam, Singapore and Algeciras have large foreign dependencies, with Rotterdam and Singapore being positioned in the middle of supply-chains (mainly due to their role as petrochemicals hub) and Algeciras more towards the end of supply-chains (given its transhipment of manufactured goods). Shanghai and Bremerhaven, on the other hand, have higher domestic dependencies and larger backward linkages. These ports are highly integrated with domestic manufacturing supply-chains (e.g. car manufacturing for Bremerhaven, and electronics and other manufacturing for Shanghai). The port of Los Angeles-Long Beach has large backward linkages, illustrating that it mainly imports goods at the end of the supply-chain, while Ulsan has large forward linkages as it plays a key role in the exports of domestically produced goods (e.g. vehicles). On the left hand side of the spectrum are ports with mainly forward linkages, implying that they mainly export goods that are used in production stages downstream in the supply-chain, such as Itaqui (iron ore and grains) and Mina Al Ahmadi (oil).

The PLOC metrics illustrate how domestic and global supply-chains are tied to the port, and how ports are positioned differently in the global supply-chain network. Although beyond the scope of this work, this measure could help evaluate the potential losses within supply-chains networks if ports are disrupted by a shock. Moreover it could help allocate maritime emissions embedded in freight flows going through ports to specific supply-chains.

## Port-level import coefficient

As economies grow, and final demand (i.e. domestic consumption and exports) changes in absolute terms and composition, imports through ports are necessary to facilitate this. Due to an increasing fragmentation (i.e. different stages of production in different countries) and globalisation (i.e. global expansion) of supply-chains[27,56], the reliance on maritime imports to support final demand has increased. As a complementary metric to describe the feedback between ports and the economy, we use the extended MRIO table to estimate the direct and indirect (through interindustry dependencies) imports per port needed to produce the domestic consumption and exports in the economies they serve. The port-level import coefficient (PLIC, see Methods) quantifies the marginal change in port-level imports for every 1000 USD change in final demand across all economies.

Figure 6a highlights the 15 ports with the largest PLIC values. These top-15 ports all have PLIC values of >170 (up to 486), with 27 ports having a PLIC of >100. The ports with the largest PLIC values are relatively small ports serving island nations (e.g. Maldives, Aruba, Mauritius, French Polynesia), but also the port of Dar Es Salaam serving demand in Tanzania and the landlocked African hinterland. Some larger ports that function as important transhipment hubs (Singapore, Kingston, Marsaxlokk and Freeport) also have large PLIC values, indicating that they are not only essential for connecting ports across the region, but also to meet the final demand in their island economies.

Similar as with the cross-border throughput dependencies, some ports are more sensitive to demand changes in foreign economies than

their domestic economy (Supplementary Fig. 9). For instance, some key ports in Africa (Djibouti, Berbera, Cotonou, Maputo) are more sensitive to changes in foreign demand than domestic demand, as they serve landlocked economies that are larger than their own. Similarly, in Europe, large foreign demand sensitivities are found for the ports of Bar (Montenegro) and Burgas (Bulgaria).

In general, larger PLIC values are found for ports in countries that have a limited number of importing ports and have a high overall trade openness, i.e. they rely disproportionally on foreign products to meet their domestic consumption and for use in domestic production processes that are later exported to other countries. To further explore the differences between countries, we aggregate the PLIC values to the economies they serve (country-level import coefficient, CLIC), indicating the USD increase in country-wide maritime imports due to a 1000 USD increase in final demand.

On a country-level, for every 1000 USD increase in final demand, ports that serve that country experience a median (maximum) 84.6 (501.5) USD increase in maritime imports, underlining large differences between countries. SIDS have a 1.5 times higher CLIC compared to non-SIDS countries (Fig. 6b). Figure 6c displays the CLIC across income groups, showing that low income countries have lower CLIC, as they are often less integrated and diverse supply-chains. In general, manufacturing sectors have larger import coefficients, requiring more maritime imports per unit of final demand[56]. For instance, across all countries, the Agricultural (sector 1) and Mining and Quarrying (sector 3) sectors require on average 40 USD for every 1000 USD change in sectoral demand, while some manufacturing sectors (sector 9 – 11) require on average 112 – 153 USD for every 1000 USD change in sectoral demand. Therefore, given that high-income countries are generally more diversified (e.g. higher manufacturing base) and better integrated within global supply-chains, they require more maritime import per USD change in final demand.

The import coefficients (on a port and country level) help to understand how future trade flows through ports will change as countries develop (e.g. demand growth), supply-chains restructure (e.g. better supply-chain integration), and sector composition shifts (e.g. higher manufacturing base).

## Discussion

This study presents a comprehensively global analysis of the different dimensions of the criticality of 1300 individual ports for the international trade, maritime transport and global supply-chain networks. The research is a significant step beyond conventional input–output analysis, which does not resolve the role of individual ports, and maritime network analysis, which does not reflect the sector-specific volumes of goods transported on the network, thereby providing a misleading prioritisation of ports' criticality. Altogether, this work present a new quantitative framework that allows one to rethink the role of specific ports in the domestic and global economy, as well as the cross-border dependencies on maritime infrastructure.

We find that the approximately 50% of global trade by value is via maritime transport, although higher values are found for the Mining and Quarrying (76%) sector. Maritime trade flows are highly concentrated in a small number of ports that benefit from economies of scale and are well-integrated with the maritime and hinterland networks. Around 50 ports (out of the 1380 considered) are responsible for 50% of global maritime trade, with this trade unevenness being much larger for certain sectors such as the manufacturing of Textiles and Wearing Apparel and Transport Equipment.

Low income economies and SIDS depend disproportionally on their port infrastructure for trade. Low income countries import 1.5 times more by means of maritime transport than high-income countries, while SIDS have a twice as high maritime import dependency compared to non-SIDS. Therefore, investments in reliable port infrastructure in low income countries and SIDS are essential if further

economic growth is not to be inhibited by port capacity[57]. The benefits of increasing trade facilitation provided by ports may reach beyond the port boundaries, as ports tend to attract industry clusters[58,59] and lower transaction costs in trade, which could lead to indirect benefits through access to international markets (e.g. food availability, expending exports markets)[60–62].

We find large cross-border dependencies between ports and the economies they serve due to land connections or transhipment services. Globally, transhipment services and the use of ports in foreign (land-connected) countries contribute to 35% of global port throughput. We identify important cross-border links between landlocked countries in Africa and South America and specific coastal ports, as well as island economies in the Pacific and the Caribbean that rely on regional transhipment hubs. The mutual dependency of economies on foreign maritime infrastructure means that there are potential spillovers when shocks or structural changes occur to either the economy or the maritime network. For instance, strong economic growth in landlocked economies or improved cross-border transport networks between landlocked countries and its maritime neighbours (e.g. the Belt and Road Initiative and the Bioceanic Road Corridor) can lead to increasing demand at the connected ports.

Port are further found to be essential to integrate domestic and global supply-chains. In relative terms, every USD flowing through a port contributes on average 4.3 USD in value to the economy. While some of the world's largest ports are found to be critical for the global economy (>1.4% of global output depends on trade going through these ports), we identify a number of ports (40) in trade-dependent economies that are critical for >10% of domestic industry output. The position of ports within supply-chains depends on the relative importance of domestic versus foreign and forward versus backward supply-chain linkages. Similar ports in terms of size may be found at different ends of the spectrum, which has important implications for the feedback between the economy and trade flows through ports, and for evaluating the potential magnitude and spatial extent of supply-chains losses if ports are disrupted.

Finally, we find that for every 1000 USD increase in final demand (e.g. domestic consumption and exports) in an economy, the ports serving that economy experience a median 85 USD increase in maritime imports. However, some (27) ports import over 100 USD per 1000 USD change in final demand in the economies they serve, most of which are ports serving small island economies, but also ports serving landlocked economies (e.g. Dar Es Salaam, Djibouti). While the maritime import requirement per USD demand change is lower for low income countries than high income countries, the import sensitivity of low income countries is expected to increase as economies grow, become more diversified and better integrated in global supply-chains.

Our quantitative modelling framework paves the way for future research in various disciplines. First, our disaggregated analysis of global trade flow could allow estimating the carbon emissions embedded into maritime transportation and can help allocate these emissions to countries and sectors[23,63]. Second, by incorporating various transport policies into the model, such as infrastructure investments (e.g. new transport routes), improved trade facilitation (e.g. reducing transit times at borders) or a (maritime) carbon tax, the changing allocation of freight flows could be evaluated. Third, by analysing future trade flows, the current analysis could help quantify the future investment needs in terms of new port infrastructure. Finally, by coupling this framework to a disaster impact model[64], the economic-wide losses (domestic and global) from port or maritime transport disruptions could be assessed, including the future losses due to climate change (e.g. sea-level rise).

In conclusion, ports are closely tied to the economy by facilitating trade flows that connect global supply-chains networks. Our research emphasises the need to rethink the key distinctive features of ports in terms of their criticality for the domestic and global economy, which

are largely hidden in aggregate port-level trade statistics. We further highlight the need to integrate long-term planning of port infrastructure with a system-wide understand of the interconnected transport and the economic system. Given the large societal dependencies on maritime transport, evaluating the key links, feedbacks and dependencies between ports and the economy is imperative for the sustainable development of economies.

## Methods
### Overview
We describe the methodology of the modal split model, the maritime transport model and the link between ports and supply-chains using the MRIO tables. Throughout the analysis, we use national economies as the spatial-level of aggregation, which we further disaggregate to the port-level. This because the international trade data and global supply-chain database are constructed on a country by country basis, restricting using subnational economic data. We do recognise that this might bias some of the results as some interpretations of the results might be related to the size of the economy. Further, throughout this research, ports are defined as one or multiple terminals within a specified port boundary, which have been delineated in line with the World Port Index, the most widely used database of ports.

### Modal split model
We develop a global modal split model to predict the share of maritime trade in every bilateral trade flow on a commodity level. A detailed description on the model is included in Supplementary Note 1. A model choice model intends to predict the allocation of freight transport flows for a given Origin-Destination (O-D) over alternative and competing transport modes[65]. Transporting goods between every O-D using a certain mode has a given utility that the shipper intends to maximize, which includes mode-specific variables (cost, time), O-D specific variables (income, neighbouring countries), and commodity specific variables (quantity, value to weight ratio, perishable). We fit the modal split model based on reported modal split data in international trade from UN Comtrade[66]. We use this model to predict the modal split in every bilateral trade flow reported in the harmonized BACI trade database[67]. This database contains over 8 million trade flows on a commodity level, which we aggregate to a 11 sector classification system we adopt throughout this work (see Supplementary Table 1). This sector classification corresponds to the 11 commodity sectors included in the EORA MRIO table, which we used later on (see Link to input−output tables). This country-to-country maritime trade database is used to model the supply and demand of goods across countries globally that are consequently allocated on the maritime transport network. An external validation of the model results is included in Supplementary Note 2.

### Oxford maritime transport model
The new global maritime transport model developed for this study (the Oxford Maritime Transport model, or OxMarTrans), combines a top-down representation of transport demand (driven by predicted maritime trade flows) with a bottom-up (asset-level) representation of the maritime and hinterland transport network. Its main purpose is to accurately allocate trade flows between countries, which we disaggregate to administrative regions within countries, on the maritime transport network, taking into consideration the likely ports and maritime routes taken based on observed sector-specific capacities between 1380 ports from empirical vessel movement data (Automatic Identification System, or AIS). A detailed model description and validation is included in Supplementary Note 3.

To the best of our knowledge, the OxMarTrans is the most detailed global maritime transport model available. It builds upon previously developed maritime transport flow models, either specifically for container flows[5,68] or multiple vessel types[41], that are used to allocate trade between countries on the maritime transport network.

However, the OxMarTrans model makes some noticeable improvements to those earlier models. First, it simulates flows between around 3400 subnational regions globally, instead of using country centroids, which better captures how different ports facilitate trade of specific hinterlands. Second, it include a multi-model hinterland transport network, which therefore captures how the port choice is driven by a better integration (better connectivity or availability of alternative modes) of ports within their respective hinterlands. Third, we embed an observed maritime transport network, based on actual vessel movements into the model, which therefore takes revealed route preferences (e.g. strategic route decisions) into consideration. Previous work has not included this, making it hard to realistically model route choices, in particular transhipment flows. Fourth, we add sector specific constraints to the model framework, helping us to capture the specialisation of different ports and, hence, the specific cargo they handle. Fifth, we perform a flow allocation per economic sector, the output of which provides an explicit link with a MRIO, which has not be done in earlier model frameworks.

The model output captures, per origin and destination country and economic sector, the share of maritime trade going through specific ports, in terms of the points of exports, transhipment and imports, as well as the maritime and land-route route taken. In this way, we can analyse both the trade flows on a port-level and the use of certain transport routes (e.g. Suez and Panama canal or hinterland transport corridor) to trade goods between country pairs.

### Link to input−output tables
To connect the port-level trade flows to an I-O table, we use the latest EORA MRIO[32] (2015 at the time of writing), which describe the intercountry and interindustry dependencies for 190 economies. Of the 207 countries included in the port-to-port trade network, 176 countries are included in the MRIO, leaving us with 1300 ports for the analysis. Trade flows included in the MRIO table are not always similar as those included in the BACI trade database[67], and hence we can only modify overlapping trade flows for this analysis (since we only derive maritime percentages for these specific trade flows).

The import coefficient is derived in line with the work of Hummels et al[56]., that used the concept of import coefficients to quantify the amount of imports embedded in the export of a country (i.e. vertical specialisation). Although the methodology of Hummels et al[56]. was developed for a single country I-O table, Dietzenbacher[69] showed that the same result holds for a MRIO. Our port-level import coefficient (PLIC) metric quantifies the amount of imports through a port ($p$) that serve as a country ($k$) that are embedded in exports ($e$, vector of exports) and domestic final consumption ($c$, vector of consumption). In a MRIO table, the input coefficients matrix ($A$) for country is derived from its interindustry trade ($Z$) and industry output ($x$). For a country k = 1, this consists of $A^{11} = Z^{11}(\hat{x})^{-1}$ for domestically produced inputs and $A^{k1} = Z^{k1}(\hat{x})^{-1}$ for inputs imported from country $k$ $(k \neq 1)$.

The domestic output necessary for $e$ is $(I-A^{11})^{-1}e$ and for $c$ is $(I-A^{11})^{-1}c$, which require imports $M = \sum_{c=2}^{k} A^{c1}$ (c=2 to k means input from other countries). Hence, the total imports to meet $e$ is $s\prime M(I-A^{11})^{-1}e$ and to meet $c$ is $s\prime M(I-A^{11})^{-1}c$, with $s$ a summation vector. To find imported goods going through a port, we modify the $M$ matrix using the port-to-port trade network, by first making $M=0$, and filling the $M$ matrix with the fraction of country-to-country trade (share time trade flow) that goes through a port per sector ($s$) (with $A_p^{c1}$ the port-level imports from country $c$ to country 1 to port $p$). This results in a new $M_p$ per port that covers the input coefficients from country $k$ to the host country of the port (country $c = 1$), which are being transported through this port. Using this, we can find the PLIC metrics by

$$PLIC_{dom,p,c} = \frac{s\prime M_p (I - A_p{}^{11})^{-1} c_c}{s\prime c_c} \qquad (1)$$

and

$$PLIC_{\exp,p,c} = \frac{s' M_p \left(I - A_p{}^{11}\right)^{-1} e_c}{s' e_c} \qquad (2)$$

Describing the port-level imports required to meet the final demand for the country the ports serve ($PLIC_p = PLIC_{dom,p} + PLIC_{exp,p}$). The total import multiplier for a country (CLIC) is found by aggregating the PLIC-measures per port that serve the demand of a country ($p,c$) ($CLIC = \sum PLIC_{p,c}$). The sector-specific import multipliers on a country-level are found by replacing $c$ and $e$ with a vector with a 1 for the specific sector and a zero otherwise.

The port-level output coefficient (PLOC) metric is a variation of the Hypothetical Extraction Method (HEM)[70–72] used in I-O analysis, in which a sector is hypothetically set to zero (the i-th row and j-th column of matrix $A$) in order to evaluate the inter-industry dependencies and importance for the economy through changes in the industry output. For the PLOC, we quantify the output changes to the economy by removing the trade flows going through a port from the I-O table. To do this, we use both supply-driven (Ghosh) and demand-driven (Leontief) versions of the I-O table to find the forward (supply-driven) and backward (demand-driven) linkages. Using a Ghoshian model is justified here as we look at reductions in industry output (see Rose and Wei for a discussion[25]). The PLOC metric is derived by (1) modifying the interindustry trade matrix ($Z$) and (2) the final demand matrix ($y$) to account for the trade flows going through a port. First, we remove the port-level trade flows (both import and export) from $Z$ and re-evaluate the new $A_{p,1}$ using the demand-driven model and the new $B_{p,1}$ using the supply-driven model ($B_{p,1} = \hat{x}^{-1} Z$). We find the backward losses in industry output ($\Delta x_{p,1,ind,b}$) by re-calculating industry output ($x_{p,1,ind}$) with the modified direct requirement matrix:

$$\Delta x_{p,1,ind,b} = x - \left(I - A_{p,1}\right)^{-1} y \qquad (3)$$

And the new industry output for the forward linkages ($\Delta x_{p,1,ind,f}$):

$$\Delta x_{p,1,ind,f} = x - v \left(I - B_{p,1}\right)^{-1} \qquad (4)$$

with $v$ the vector of value-added. The changes in industry output is the addition of the changes in domestic output ($\Delta x_{dom,ind,b}; \Delta x_{dom,ind,f}$) and change in output on the rest of the economy ($\Delta x_{row,ind,b}; \Delta x_{row,ind,f}$). Moreover, we evaluate the changes in industry output due to the port-level trade embedded in direct consumption. This is done by modifying the demand matrix ($y$) with the equivalent reduction in domestic final consumption (imports) and the reduction in final consumption in other countries (exports). Output losses associated with changes in final consumption in a port in country 1 ($y_{p,1}$) can be found by solving:

$$\Delta x_{p,1,con,b} = x - (I - A)^{-1} \left(y - y_{p,1}\right) \qquad (5)$$

From $\Delta x_{p,1,con,b}$ we can find changes in domestic output ($\Delta x_{dom,con,b}$) and changes in output for the rest of the economy ($\Delta x_{row,con,b}$) in a similar fashion as described above. The forward losses associated with trade in final consumption are simply the trade flows of final consumption, with imports leading to a reduction of domestic consumption ($\Delta c_{dom,con,f}$) and exports leading to a reduction in foreign consumption ($\Delta c_{row,con,f}$).

This yields the PLOCA metric, which can be derived from changes in output and consumption:

$$PLOCA = (\Delta x_{dom,ind,b} + \Delta x_{dom,con,b}) + (\Delta x_{dom,con,f} + \Delta x_{dom,ind,f})$$
$$+ (\Delta x_{row,ind,b} + \Delta x_{row,con,b}) + (\Delta c_{row,con,f} + \Delta c_{row,ind,f}) \qquad (6)$$

from which PLOCR can be derived:

$$PLOCR = \frac{PLOCA}{T} \qquad (7)$$

With $T$ the throughput going through the port. The relative importance of the global versus domestic economic linkages and forward versus backward economic linkages can be derived by dividing the different components of PLOCA (neglecting the final consumption changes). The global and domestic importance of ports is simply derived by dividing the total industry output changes with the global and domestic industry output. A similar exercise is done for the final consumption changes.

### Reporting summary
Further information on research design is available in the Nature Research Reporting Summary linked to this article.

### Data availability
All data to reproduce the results of this study are available on Mendeley Data (https://data.mendeley.com/datasets/vzzy3b9gg4/1).

### Code availability
The code to run the maritime transport model and quantify the link between ports and supply-chains is available on GitHub (github.com/jasperverschuur/Port_supply_chains). All other codes could be requested from the corresponding author upon reasonable request.

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

## Acknowledgements

The authors would like to thank the United Nations Statistical Division and the UN Global Working Group on Big Data for Official Statistics, in particular Markie Muryawan and Ronald Jansen, for providing the mode of transport data and the AIS data. Moreover, we like to thank Lóri Tavasszy and Luis Martinez for discussions and data provision that helped improve the methodology. J.V. acknowledges funding from the Engineering and Physical Sciences Research Council (EPSRC) under grant number EP/R513295/1. E.E.K. was further supported by the Netherlands Organization for Scientific Research NWO (Grant No VI.Veni.194.033).

## Author contributions

J.V., E.E.K. and J.W.H. designed the study. J.V. performed the analysis with input from E.E.K. and J.W.H., J.V., E.E.K., J.W.H. contributed to the writing and reviewing of the paper.

## Competing interests

The authors declare no competing interests.
