## [Peer Review File · Nature Communications]

Reviewer comments, first round review: –

Reviewer #1 (Remarks to the Author):

Major points

-the article does not mention or use the widely made distinction between container liner shipping and bulk tramp shipping. However, this distinction may be relevant. For instance, container ships operate in services with a large number of intermediate stops. Between Rotterdam and Shanghai, around 10 intermediate ports are called. This is different from bulk, where there are hardly intermediate stops (i.e. a ship will be fully loaded in 1 port and fully unloaded in another port). I wonder whether ignoring this difference impacts the quantitative methods, more specifically the approach to trade distribution. It would seem that distributing trade flows of coal (bulk) and machinery (containers) to ports with a model that does not take the differences in maritime networks into account is problematic.

I wonder if the same applies to the core periphery method. While in container networks, links with intermediate stops in between constitute real connections (in the sense that flows use them) that is not (or much less) the case in bulk shipping. Not taking this into account may lead to the finding that Port Hedland has a higher coreness than say Melbourne. Such a finding is a bit counterintuitive when associating coreness with connectivity.

Finally, I also wonder if the bulk/container distinction impacts the approach to diversification. As containers are used to transport different commodities (machinery, textiles, food, paper, ...), container ports are likely to be more diversified than bulk ports. The value density of goods carried in containers is also much larger than that of bulk goods. Is that part of the explanation of the relation between diversification and coreness?

-The article mentions forecasts multiple times, without acknowledging the consequences of disruptions for future trade flows. The outlook referred to (line 31) is not really focused on maritime transport, this study includes scenarios with much less volume growth due to disruptions like 3D printing and more circularity. Another study, from DNV GL focused on the energy transition yields different forecasts (but this study also is not really based on a deep understanding of maritime trade and its drivers). The value of the import coefficients for assessing the future growth of ports is in my view limited, all kinds of specifics deeply impact the development of volumes. Just as an example: port volumes in the UK have decreased in the last decades, while output has increased.

-I doubt whether the way the PLOC is constructed makes sense. If I understand the method, the trade flows through a port are removed from an I-O table. That seems to assume that there are no alternative ports that could accommodate these trade flows? The metric may say more about the dependence of a country on maritime trade than on a specific port. For instance, the way 'dependent' is used line 15 seems to stretch the findings.

Minor points

-on the logic of the selected commodities: some more explanation may be helpful. Have you considered excluding S2 Fishing? Seems to have specific issues (e.g. fishing by a foreign fleet in national waters of a third country etc).

-the abstract can be improved in terms of style

-I am certainly not an expert but was wondering if the terms 'downscaling' (line 12 and elsewhere) and 'embeddedness' (line 16 and elsewhere) are used in their conventional meaning?

-line 39 ref 8: I am not sure the statement accurately reflects the cited paper.

-line 124 the logic of income levels and the ability to afford air transport is not so convincing, this seems to be more related to supply chain characteristics than income levels (see flower exports from Tanzania and Ethiopia).

-line 137-139: there may not be an overreliance; the logic may go in the opposite direction, limited integration explains the higher share of maritime transport; i.e. it will disappear 'spontaneously' once the export flows change.

-line 222: ignores the possibility of transit cargoes

-there are some inevitable simplifications such as the capital city and the maritime distances.

Would be good to understand their impact on the findings. For some countries, the impact seems

relevant (e.g. Russia, Canada, US, Colombia, Mexico).

Line 581: perhaps explain in more detail the underlying logic?

F1: the bar chart in the figure is a bit hard to follow

Reviewer #2 (Remarks to the Author):

The manuscript under review reports on an ambitious project to link trade flows to specific ports. This is a welcome addition to our knowledge given that much of the trade literature seems to ignore that trade in goods 'takes place' and materializes in the form of traffic that makes use of infrastructure. Moreover, it adds to the literature on ports and maritime transport, where the link with economic activity in the hinterlands of ports is usually discussed with less detail. It is promising that the approach goes beyond case studies, which are also abundant in the field of Global Value Chains, and aims to cover the entire world and all relevant industry sectors. The core of the work is based on a multi-region input-output table, the Eora MRIO. A first step is to remove non-maritime trade flows by estimating the share of each flow that is transported by ship, air and road, using a modal split model. Subsequently, the maritime country-to-country trade flows are attributed to port pairs on the basis of a detailed database of actual vessel traffic. The resulting database contains disaggregated data on 157 countries, 1132 ports and 11 industry sectors. Note that a variety of data sources is used throughout the paper, not just the MRIO table. Finally, the data is applied to estimate (1) the share of maritime transport in global trade, (2) the share of trade (disaggregated per sector) that passes through specific ports, (3) the degree of specialisation and the coreness of ports, (4) the port-level output coefficient (PLOC), 'that captures the total industry output directly or indirectly dependent on the trade flows through a port', and (5) the port-level import coefficient (PLIC) which gives 'the direct and indirect (through interindustry dependencies) imports per port needed to produce the domestic consumption and (re-)exports in an economy'. My comments relate to (1) the aims of the paper, (2) the conceptual framework, and (3) empirical challenges.

Aims of the paper

The abstract indicates that the aim of the manuscript is twofold. On the one hand, it offers 'a comprehensive methodology to downscale global trade flows to a port-level, which allows us to incorporate all significant ports in a global supply-chain database', on the other hand, the aim is to evaluate the criticality of ports for the economy, and more specifically, to obtain a 'better understanding [of] the feedback and dependencies between ports and the economy, thereby supporting long-term resilient infrastructure planning'.

The first aim is to develop a methodology to obtain a new type of database. Regarding this aim, surprisingly little is said about existing databases. For quite some time, statistical agencies, consultancies as well as other companies provide data on trade flows, port throughput, and the economic structure of countries and regions. As a consequence, the obvious question is, why has no-one come up with this or a similar dataset before? One reason might be that many sources are behind paywalls. The current data can be presented as a freely available alternative to products from commercial players such as Clarksons, Containerisation International, IHS Global Insight or Seabury. Another possible reason is that we had to wait until the development of global supply chain databases such as the Eora MRIO. I leave aside the possibility that the reason for the current absence of such a database is its irrelevance. In other words, the article needs a clear position relative to the state-of-the-art. Reference is made to studies in different relevant fields including Global Value Chains, port studies and MRIO, but what is lacking is an explanation why the work reported here is presented today, and was not done before. My guess is that there is more at stake than the empirical challenge to link all the different data sources. The manuscript under review is based on a paradigm that is not mainstream in (trans)port economics, to say the least. Port economics is almost a synonym for neoclassical port economics and the promotion of marginal cost pricing, and the field has a legacy of fighting protectionist tendencies of governments that want to promote their own, national ports.¹ It is known that neoclassical economists often see input-output analysis as an instrument of the central economic planning they oppose, while input-output economists argue that neoclassical economists ignore the structure of the economy.² Seen from that perspective, the aim of the paper is not just to introduce a dataset or to estimate the

impact of disruptions, but to import a different paradigm. Note that examples of input-output analysis can be found in the literature on ports, but mainly to investigate the port sector as an industry.³⁻⁵

The second aim, as the title suggests, is to provide insights in the criticality of ports in international trade and global supply-chains. If I'm correct, 'criticality' is used as a synonym for resilience,⁶ however, 'Most analysts use resilience in a common sense and non-rigorous fashion'.⁷ Assuming that the five aforementioned applications reveal what the authors mean by criticality or resilience (and port embeddedness), the definition of the concept remains vague when looking at these five interpretations.

(1) the share of maritime transport in global trade seems to be used as a crude way to stress the importance of maritime trade, reminiscent of the references made to UNCTAD's Reviews of Maritime Transport:

'For an economic region such as the European Union, shipping accounts for 80% of total exports and imports by volume, and some 50% by value.'⁸

'Maritime shipping is the backbone of world trade; it is estimated that some 80 percent of all goods are carried by sea. In terms of value, global maritime container trade is estimated to account for around 60 percent of all seaborne trade, which was valued at around 14 trillion U.S. dollars in 2019.'⁹

'According to the United Nations Conference on Trade and Development (UNCTAD), more than 80% of global trade by volume, and more than 70% by value, travels by ship. UNCTAD put the volume of seaborne trade in 2015 at 10.3 billion tons, with a combined value of \$25.3 trillion.'¹⁰

Given the rigorous and academic nature of the work, I expected a note on how the methodology differs from that of, in the first place, UNCTAD, on whether the finding that '55% of global trade (by value) is maritime' corroborates existing estimations (as a kind of triangulation), and on the meaning of these figures (in what sense do these measure 'criticality?'). Similarly, the fact that low-income countries have a lower share of aviation, export relatively more mining products, often depend on the export of a limited number of commodities, and are less well integrated in global shipping networks (see also (3)) is an established fact. This does not reduce the relevance of the observation, but requires that it is presented as corroboration of existing information. Finally, the conclusion that 'On average, every 1000 dollar increase in final demand of an economy leads to an additional 2.2 dollar of imports going through a given port' and that 'On a country-level, for every 1000 dollar increase in final demand, ports (i.e. all ports in a country) will experience a mean (maximum) 18.3 (108.3) dollar increase in maritime imports.', might also be framed as a critique of claims that maritime trade is crucial (2.2/1000 does not sound impressive). Some additional interpretation might be required.

(2) the second measure is the share of trade that passes through specific ports (e.g. '9.3% of the total industry output depends upon trade flowing through only the top 10 ports'). Here the focus is on ports, and a ranking of ports is given.

(3) also the third interpretation of criticality takes a port's view and is measured as the degree of specialisation and the coreness of a port. This interpretation follows the logic of network analysis (and graph theory). Here too, reference can be made to existing measures (e.g. UNCTAD's liner shipping connectivity index).

(4) the port-level output coefficient (PLOC), is based on input-output analysis,

(5) the port-level import coefficient (PLIC), is like the PLOC, based on I-O analysis.

Although the two last coefficients are measured at the port-level, the emphasis is on the country-level (the 'economy', and are closely related to country-level import coefficients (CLIC)). Hence, measures of 'criticality/resilience' are found for different things, which leads to the recommendation to be clear of what the 'criticality' is estimated.

A third (potential) aim, contributing to environmental research and policy, is only briefly mentioned (e.g. the data can help to 'connect environmental footprints with commodity flows', 'estimating the carbon emissions embedded into maritime transportation' and 'maritime

decarbonisation strategies'). Nevertheless, besides the affiliation of the authors and the scope of the journal ('research from all areas of the natural sciences'), the MRIO-framework fits well with environmental aims.¹¹ Surprisingly, no critical remarks with regard to environmental impacts were made about the observation that 'Future demand scenarios project a near tripling of maritime trade volume by 2050' (besides, the first paragraph of the paper looks as if it was copied from promotional material of a port, in contrast to the rest of the article). However, taking the environmental road might imply different methodological choices, like focusing more on weight than on value. And what about the concept of international trade itself? From a transport and environment perspective, I'm interested in ton-kilometres, vehicle technology and modal split, not in whether goods cross national borders. I recommend to mention whether environmental impacts fall outside the scope of the paper, or alternatively, can easily be included in the methodology.

Conceptual framework

An interesting feature of the manuscript is that it combines insights, concepts and data from different fields, including MRIO analysis, port and maritime studies, and the global value chains approach. However, this might come at the cost of conceptual clarity.

Many concepts are rather clearly defined in the article (e.g. (dominant) transport mode, PLOC, PLIC,...). As was discussed earlier, some confusion might rise regarding the exact definition of criticality and resilience (and whether this is the same as port embeddedness).

Taking into account the background of the target audience of the journal, it might be advisable to make clear what the 'economic' concepts exactly mean. I recommend to make clear how the 'world' is approached, i.e. using the MRIO framework. If someone applies a gravity model, the idea behind the model is pretty easy to explain, i.e. the amount of interaction (trade) between two regions depends on the size of the economies and the distance. Something similar can be done for MRIO. A picture (flowchart) can be used to explain the essence of the chosen framework, which needs to be accompanied by an argumentation why this view on the world is the most appropriate. Discussions of 'Trade in Value Added' illustrate that even the concept of 'value' might be measured and interpreted in various ways.¹² A simple diagram might help readers to understand what concepts mean and how these relate to each other, think of (total) industry output, intermediate/final demand, direct/indirect imports, exports, direct/final/domestic consumption, forward/backward linkages, feedback,... Specifically, the concepts of dependency (and influence) need careful introduction as these can be interpreted rather mechanically, and can find easily their way to newspapers and other media (for example, 'In total, 41.9% of global industry output depends on the trade flows going through port').

Another relevant concept is 'losses'. When it comes to the use of the results for policy, the article aims to contribute to a better understanding of 'how demand shocks (e.g. pandemic) ripple through the economy' and of 'potential losses within supply-chains networks if ports are disrupted by a trade shock (e.g. natural disaster or labour strike).' To this end the paper mobilises 'the port-level output coefficient (PLOC) that captures to what extent industry output is reduced (both domestically and globally) if a particular port is hypothetically extracted from the I-O table' (short-term), and 'the port-level import coefficient (PLIC) that measures the increase or decrease in port-level trade flows when demand changes in an economy' (longer-term). Moreover, 'by coupling this framework to a disaster impact model, the economic-wide losses (domestic and global) from port disruptions could be assessed, including the future losses due to climate change (e.g. sea-level rise, increased frequency of flooding)'. The approach could also 'help refine the future investment needs of port infrastructure and evaluate the changing dependencies between ports and the economy' (long-term infrastructure planning). To have an idea of the short-term impact of disruption, I-O based approaches can give an indication of the industries that will be affected by the disruption, together with an order of magnitude. As has been discussed in the literature, in the short term, substitution of one input by another is less likely, just like major reorganizations in production. When disruptions last longer, the structure of I-O relations will probably change, which makes it less meaningful to hypothetically extract a port from the I-O table.⁶ For the longer term (and not just for reductions in industry output, but also increases), it is hypothesized that an increase in demand has an impact on port throughput, without taking into account that changes in ports (that affect the generalized transport cost) have a fundamental impact on the structure of the economy. The latter is very common, and decreasing transport costs (associated with investments in infrastructures such as ports) are seen as the driver of globalization and as a force that changes the structure of economies and the spatial pattern of economic activity (among others in the New Economic Geography literature¹³⁻¹⁵). From a neoclassical perspective, the

analysis is presumably seen as too static, and too mechanical, to say something meaningful about the effects in the longer term. But this is a matter of choice of paradigm, and as long as someone is open about the adopted paradigm, and aware of the shortcomings and criticisms, choice of paradigm is not in itself a basis to reject a particular approach. What can be criticized is that the current version of the manuscript is not so clear about the importance of paradigmatic differences. From a port economics perspective, this is not simply an empirical exercise which leads to a new dataset, but a different paradigm that is applied (see discussion above). Turning back to the policy-oriented applications presented in the manuscript, a key concept is 'economic losses' which are caused by disruptions. The idea of a loss presupposes a reference base, usually the situation just before the disruption appeared, and a counterfactual, i.e. what would happen if the disruptive event would not take place (business as usual). Natural disasters (including floodings caused by climate change) and labour strikes serve as examples of disruptions. However, what about the corresponding counterfactuals (the *ceteris paribus* question)? Someone who builds a port in a hurricane-prone region should know that there will be a certain number of days that port operations will be 'disturbed' by bad weather. It's hard to predict the exact number of hurricane days in a given year, but the counterfactual of a port in a hurricane-prone region without hurricanes is a contradiction in terms. When 'natural' disasters are caused by human economic activities, as is the case with climate change, then the counterfactual of an identical economic structure (I-O table) without greenhouse gas emissions is also not very likely. The same can be said about labour strikes. In a perfect neoclassical market, wages reflect the value of a person's labour output. When wages are lower than this value, this might induce a strike. Low wages in ports are beneficial for the industries that depend on them since transport costs are lower. The counterfactual of low wages without any action from the labour side is, again, unlikely. We can continue the long-lasting debate on whether monopolies on dock labour distort markets, or whether it's more of a monopsony, but the point is that many strikes are not something external to an economy. I got the impression that the concept of economic loss is linked to external disruptions. What needs to be explained, then, is what a disruption and the resulting economic loss actually are.

Empirical choices and issues

The methodology in the manuscript is presented as a 'spatially explicit analysis', and from an international trade analysis perspective, this is a step forward. However, the definition of 'economies' is rather traditional as 'economies' are equated with national states. Paul Krugman once asked the question 'Why is the process that puts a bottle of French wine on a Berlin table very different from that which puts California wine on New York tables?' (p.49).¹⁶ In this respect, following remark in the manuscript is relevant: 'European countries have a much lower share of maritime transport, mainly due to the large flows between European countries'. The spatial aggregation of economic data might have a major impact on some of the results, as has been discussed elsewhere for MRIO analysis.¹⁷ Especially for large countries such as the US and China this might matter, as it does for integrated trade blocks and custom unions such as the EU. The use of capital cities as the location of airports might also cause some bias (cf. 'the geographical distance between capital cities'; 'the fact that we use capital cities as centroids, thereby estimating higher transport costs for road transport'). Also the inclusion of a large amount of small island states without weighing factors can bias some findings. For some, this is not only a technical issue related to the division of the world in arbitrary spatial units (~ Modifiable Areal Unit Problem),¹⁸ but also a substantive issue as 'national economies' do not exist according to among others World City scholars who found inspiration in the work of Jane Jacobs, and developed a framework to use cities as unit of analysis.^{19,20} I suggest to note the problem with spatial aggregation somewhere in the text, and to check what the impact might be.

The same holds for ports, which are seen as nodes in value chains.²¹ But here too, the delimitation of port areas is not without its challenges. The territory of some port areas is located in two countries (Copenhagen-Malmö; North Sea Port), or includes locations separated by a large distance. If the French government would decide to make of the Port of Le Havre-Antifer a separate port, this would be a highly specialized port (oil). Hence the question, to what extent does the 'arbitrary' aggregation of terminals in ports affects the results? Some even argue that it is rather the terminal than the port that has become the relevant unit of analysis.²² Alternatively, one can aggregate data in port ranges (e.g. the Hamburg-Le Havre range),²³ and 'port regionalization' has been widely debated in the ports literature.²⁴ I recommend to add a note on the definition of 'port' applied in the paper, and a reflection on the possible effects of other ways to

aggregate data.

In some cases averages are taken (e.g. 'We find a coreness value per port per sector and construct the total coreness value of the port by taking the mean of the coreness values across sectors.'). I'm rather skeptical towards taking the mean of values across sectors, given the difference in importance of sectors, and the uneven spatial distribution of sectors. Are alternative measures taken into consideration, or is some kind of sensitivity check carried out?

Small comments

-'Overall, specialised ports have lower PLIC (Spearman's rank correlation: -0.24), while the coreness positively scales with the PLIC (Spearman's rank correlation: -0.50).': Is it correct that both correlation coefficients have a negative sign?

-'For instance, Hurricane Katrina (2005) and the Hanshin-Awaji earthquake (1995), damaging the ports of New Orleans and Kobe, respectively, resulted in large shocks to maritime transport networks and global supply-chains. Moreover, the COVID-19 pandemic, disrupting supply-chains globally, reduced global maritime transport by 7.0-9.6% during the first eight months of 2020.': the recent grounding of the Ever Given in the Suez canal can be added as an example.

-'whereas Middle-Eastern and South American rely more': add 'countries' between 'American' and 'rely'

-'The large unevenness in terms of trade flows suggest a hierarchical structure of trade within the port-to-port trade network.': is 'hierarchical' here used as a synonym for 'hub and spoke', which also implies that the type of vessels differs between ports.

-'The products that are imported through a port are directly consumed in a country or are used in production processes to produce goods for domestic consumption or export.': is it possible that a product is imported and exported without being 'used in production processes'?

- 'the mode-specific transport costs per tonnes per km (US\$/t/km)': to what extent is taken into account that these values differ between countries (cf. road speeds), and especially for maritime trade, between vessels of different sizes?

-Modal choice model: if the goal is explanation and interpretation, the choice of variables is important. However, when the goal is to find a function that fits the data well, adding more variables (and multicollinearity) is not an issue. One can add, for example, a dummy variable for each origin and destination country, instead of using variables such as GDP per capita. When the modal choice model is simply a step towards a database, more variables can be added to increase model fit. Furthermore, dummy variables that represent country pairs or considering countries in the same trade block or customs union as neighbours (instead of 'neighbouring country dummies') can be considered.

-'For landlocked countries (which are not included in the port-to-port dataset), we use the road network to connect landlocked countries to the closest 15 ports in 15 countries, to allow landlocked countries to use ports in different countries (e.g. Switzerland can use the an Italian port for trade from Asia and a port in Germany for trade from Scandinavia).': If I'm correct, this procedure is only followed for landlocked countries. However, it is not ridiculous to claim that Rotterdam and Antwerp are 'German' ports, although these are located in the Netherlands and Belgium respectively. Especially Germany's Rhine-Ruhr region has good connections to ports in the Low Countries.

-'We validate how the well the disaggregation method': delete 'the'

-'although the methodology generally concentrates to much trade in smaller ports.': 'to' -> 'too'?

-'Trade flows included in the MRIO table are not always similar as those included in the BACI trade database': here too, some methodological interpretation and discussion might be relevant. HS is developed by the World Customs Organization (WCO), and is based on the nature of the commodity which fits well with its use by Customs. However, for economic analyses, the logic of customs is not always appropriate, and for example in SITC the end use of a good defines its classification in a category. This illustrates how conceptual and measurement issues relate to each other. Another reported issue is that customs-based import data are in general more reliable than export data since customs in the first place check goods that enter the country.²⁵ Furthermore, trade data stems from countries and different data collection practices exist in different countries.²⁶

-Although the work done is impressive, I suggest adding a warning for those who want to use data from a particular flow of goods through a particular port. When looking at the information in the supplementary files, the difference between estimated flows and actual flows can be substantial for particular cases (see 'Supplementary Fig. 9 | Validation trade allocation algorithm',

'Supplementary Table 6 | Validation of the derived maritime trade shares for European Union countries').

Conclusion

The work done is a major effort to develop a database that links port throughput to economic activity. Such a database can have many applications. In general, I recommend to:

- be more explicit about the chosen paradigm, its relation to alternative conceptualizations in the relevant literatures, and the added value of the paradigm
- make a clear distinction between the 'ideal database' (ideal from a conceptual perspective), and the actual results
- discuss the conclusions, results and methodology in light of existing findings and the state-of-the-art in the literature
- perform a kind of sensitivity analysis to estimate the impact of spatial aggregation of countries and ports

References

1. Vanoutrive, T. & Zuidhof, P. W. Port Economics in Search of an Audience: The Public Life of Marginal Social Cost Pricing for Ports, 1970-2000. *oeconomia* 9, 371–394 (2019).
2. Raa, T. T. A neoclassical analysis of total factor productivity using input-output prices. in *Wassily Leontief and Input-Output Economics* (eds. Dietzenbacher, E. & Lahr, M. L.) 151–165 (Cambridge University Press, 2004). doi:10.1017/CBO9780511493522.011.
3. Chang, Y.-T., Shin, S.-H. & Lee, P. T.-W. Economic impact of port sectors on South African economy: An input–output analysis. *Transport Policy* 35, 333–340 (2014).
4. Kwak, S.-J., Yoo, S.-H. & Chang, J.-I. The role of the maritime industry in the Korean national economy: an input–output analysis. *Marine Policy* 29, 371–383 (2005).
5. Wang, Y. & Wang, N. The role of the port industry in China's national economy: An input–output analysis. *Transport Policy* 78, 1–7 (2019).
6. Verschuur, J., Koks, E. E. & Hall, J. W. Port disruptions due to natural disasters: Insights into port and logistics resilience. *Transportation Research Part D: Transport and Environment* 85, 102393 (2020).
7. Rose, A. & Wei, D. ESTIMATING THE ECONOMIC CONSEQUENCES OF A PORT SHUTDOWN: THE SPECIAL ROLE OF RESILIENCE. *Economic Systems Research* 25, 212–232 (2013).
8. ics-shipping. Shipping and world trade: driving prosperity. <https://www.ics-shipping.org/shipping-fact/shipping-and-world-trade-driving-prosperity/>.
9. Statista Research Department. Container shipping - statistics & facts. <https://www.statista.com/topics/1367/container-shipping/> (2021).
10. World Ocean Observatory. Sea Trade: How Do We Value the Ocean? <https://medium.com/world-ocean-forum/sea-trade-how-do-we-value-the-ocean-aa1c61fff9f7> (2018).
11. Wiedmann, T., Wilting, H. C., Lenzen, M., Lutter, S. & Palm, V. Quo Vadis MRIO? Methodological, data and institutional requirements for multi-region input–output analysis. *Ecological Economics* 70, 1937–1945 (2011).
12. Johnson, R. C. & Noguera, G. Accounting for intermediates: Production sharing and trade in value added. *Journal of International Economics* 86, 224–236 (2012).
13. Fujita, M. & Mori, T. The role of ports in the making of major cities: Self-agglomeration and hub-effect. *Journal of Development Economics* 49, 93–120 (1996).
14. Thisse, J.-F. How Transport Costs Shape the Spatial Pattern of Economic Activity. (OECD-ITF Joint Transport Research Centre Discussion Paper No. 2009-13., 2009).
15. Ducruet, C., Notteboom, T. E. & De Langen, P. W. Revisiting Inter-Port Relationships under the New Economic Geography Research Framework. in *Ports in Proximity: Competition and Coordination among Adjacent Seaports* (eds. Notteboom, T. E., Ducruet, C. & De Langen, P. W.) 11–27 (Ashgate, 2009).
16. Krugman, P. Where in the World is the 'New Economic Geography'? in *The Oxford Handbook of Economic Geography* (eds. Clark, G. L., Feldman, M. P. & Gertler, M. S.) 49–60 (Oxford University Press, 2000).
17. Su, B. & Ang, B. W. Input–output analysis of CO2 emissions embodied in trade: The effects of spatial aggregation. *Ecological Economics* 70, 10–18 (2010).
18. James, R. D. & Campbell, H. S. The effects of space and scale on unconditional beta convergence: test results from the United States, 1970–2004. *GeoJournal* 78, 803–815 (2013).

19. Taylor, P. J. *World city network: a global urban analysis*. (Routledge, 2004).
20. Taylor, P. J. Being Economical with the Geography. *Environ Plan A* 33, 949–954 (2001).
21. Robinson, R. Ports as elements in value-driven chain systems: the new paradigm. *Maritime Policy & Management* 29, 241–255 (2002).
22. Olivier, D. & Slack, B. Rethinking the Port. *Environ Plan A* 38, 1409–1427 (2006).
23. Dorsser, C. V., Wolters, M. & Wee, B. V. A Very Long Term Forecast of the Port Throughput in the Le Havre – Hamburg Range up to 2100. *European Journal of Transport and Infrastructure Research* Vol 12 No 1 (2012) (2012) doi:10.18757/EJTIR.2012.12.1.2951.
24. Notteboom *, T. E. & Rodrigue, J.-P. Port regionalization: towards a new phase in port development. *Maritime Policy & Management* 32, 297–313 (2005).
25. Goldfarb, D. & Thériault, L. Canada's 'Missing' Trade With Asia. (The Conference Board of Canada, 2008).
26. UN. An overview of National Compilation and Dissemination Practices Updated Chapter 1 of International Merchandise Trade Statistics: Supplement to the Compilers Manual, United Nations publications, Sales number: 98.XVII.16,.

General Response

We would like to express our gratitude for the comments the reviewers provided. Based on these comments, we have implemented major revisions to the manuscript, which also explains the time it took to revise the manuscript to its current form. In short, the major steps taken include:

- We extended the initial number of countries considered from 177 to 207 by including landlocked countries in the analysis (see next point) and filling in some data gaps of countries initially missing from the analysis.
- We replaced the trade allocation methodology with a detailed transport model, which has the benefit that (1) landlocked countries can be included in the analysis by their dependency on foreign ports, (2) the routing on the network is included, which therefore includes transshipment and allows distinguishing between liner and tramp shipping, (3) we can add foreign infrastructure dependencies to the analysis, which was not considered in the previous method. This new approach not only allowed us to take a deeper dive into maritime transport network but also significantly improved the results.
- We removed the coreness and specialisation analysis. Based on all reviewers' comments, we think the contribution it makes to the overall paper is less than initially envisioned and henceforth we decided to remove this paragraph.
- We made some small modifications to the PLIC metric since we are now including landlocked countries and we made a small modification to the initial formulation. Initially we added an equal weighting to all sectors (e.g. how much additional maritime imports is expected when all sector increase demand with 1USD). However, in line with the original paper describing this method, and because we want to make a claim how a 1USD demand growth in the economy results in additional maritime imports, one needs to weight by the sector presence in final demand. We have implemented this now, which has increased the PLIC values.
- We made changes to the framing of the paper and its contributions in line with the reviewers' suggestions.

Reviewer #1 (Remarks to the Author):

We would like to thank the reviewer for his thorough review of the manuscript and the comments provided. The reviewer rightfully highlighted elements that needed further clarification and adjustments.

Please refer below for our detailed response.

Major points

-the article does not mention or use the widely made distinction between container liner shipping and bulk tramp shipping. However, this distinction may be relevant. For instance, container ships operate in services with a large number of intermediate stops. Between Rotterdam and Shanghai, around 10 intermediate ports are called. This is different from bulk, where there are hardly intermediate stops (i.e. a ship will be fully loaded in 1 port and fully unloaded in another port). I wonder whether ignoring this difference impacts the quantitative methods, more specifically the approach to trade distribution. It would seem that distributing trade flows of coal (bulk) and machinery (containers) to ports with a model that does not take the differences in maritime networks into account is problematic.

We thank the reviewer for pointing this out. Because we believe that this distinction is critical and should be included in our analysis, we have decided to replace the initial trade allocation method with a newly developed maritime transport model, which models the flow allocation of different commodities on the maritime network. Since we still embed the observed maritime transport network into the model, we are able to simulate transshipments as well, and therefore can distinguish between tramp and liner shipping. As we show in our results, the importance of transshipments globally (in

value terms) is quite large and we therefore believe we make a significant improvement to the model framework by including it.

I wonder if the same applies to the core periphery method. While in container networks, links with intermediate stops in between constitute real connections (in the sense that flows use them) that is not (or much less) the case in bulk shipping. Not taking this into account may lead to the finding that Port Hedland has a higher coreness than say Melbourne. Such a finding is a bit counterintuitive when associating coreness with connectivity.

We agree with this comment. We think that the core-periphery method is partly influenced by methodological choices/assumptions and in the end does not contribute as much to the overall manuscript framing as initially envisioned. Because of this comment, and other comments about the quantification of coreness and specialisation, we have decided to remove this paragraph from the final manuscript.

Finally, I also wonder if the bulk/container distinction impacts the approach to diversification. As containers are used to transport different commodities (machinery, textiles, food, paper,), container ports are likely to be more diversified than bulk ports. The value density of goods carried in containers is also much larger than that of bulk goods. Is that part of the explanation of the relation between diversification and coreness?

Yes this would explain part of this relationship. For instance, purely container ports will be more diversified than let's say a coal exporting port, even though they both specialise in one subtype of vessels. In terms of the value density this is also true but a bit more nuanced, as for instance specialised vehicle ports might still be high value.

For this reason, and the other reasons mentioned before, we decided to exclude this analysis from the manuscript.

-The article mentions forecasts multiple times, without acknowledging the consequences of disruptions for future trade flows. The outlook referred to (line 31) is not really focused on maritime transport, this study includes scenarios with much less volume growth due to disruptions like 3D printing and more circularity. Another study, from DNV GL focused on the energy transition yields different forecasts (but this study also is not really based on a deep understanding of maritime trade and its drivers). The value of the import coefficients for assessing the future growth of ports is in my view limited, all kinds of specifics deeply impact the development of volumes. Just as an example: port volumes in the UK have decreased in the last decades, while output has increased.

As the reviewer rightfully highlights, different maritime trade outlooks exist. Some forecasts are made based on time series extrapolation and not based on a deep understanding of global supply-chains and underlying drivers of the magnitude and composition of trade. The ITF outlook mentioned does include a component on demand modelling based on population and GDP growth, but, as the reviewer underlines, is also focused on evaluating a large number of policies and other disruptive trends (such as 3d printing) that can change the magnitude of transport and the modal split.

A study that, in our opinion, has performed a good scenario outlook is Walsh et al. (2019) who tried to integrate maritime trade scenarios with the different climate and socio-economic scenarios (SSPs and RCPs). This study shows a similar growth potential compared to the ITF outlook (trade in 2050 will grow two to four times compared to 2010). We therefore replaced this citation and the statement made.

On the import coefficient, it is indeed true that the present-day import coefficient might be less insightful for the long-term future, as the structure of the domestic economy, and the value-added structure of supply-chains will inevitably change. To be useful, future import coefficients needs to be

established as a result of changes in the global supply-chain structure. However, forecasting trade is one thing (and already really hard), but forecasting supply-chain structures is simply impossible. The example of the UK is striking, as the industry composition and supply-chain structure of the country has changed, likely resulting in less export of high volume bulky goods, and more import of low volume containerized goods, alongside a growth in industry output.

However, for the near-term, we do think that the import coefficients have value, as the structure of the economy will not change considerably. Moreover, we do think it has value in explaining the dependency on maritime trade for a domestic economy, which is more meaningful than just saying that maritime trade is X% of national GDP.

Based on this discussion, we have toned down the argument that the PLIC is suitable for long-term forecasting (L433-436).

C. Walsh, *et al.*, Trade and trade-offs: Shipping in changing climates. *Mar. Policy* **106** (2019).

-I doubt whether the way the PLOC is constructed makes sense. If I understand the method, the trade flows through a port are removed from an I-O table. That seems to assume that there are no alternative ports that could accommodate these trade flows? The metric may say more about the dependence of a country on maritime trade than on a specific port. For instance, the way 'dependent' is used line 15 seems to stretch the findings.

The reviewer is correct that the port is hypothetically removed from the IO table to understand its importance for the domestic and global supply-chains through all the complex intersectoral/country dependencies on this port. This methodology is in line with the Hypothetical Extraction Method (HEM) often used to understand the criticality of a sector. See for instance Dietzenbacher *et al.* (2019): "A method that has been widely used to measure interindustry linkages and the importance of industries is the hypothetical extraction method. HEM considers the hypothetical situation in which a certain industry is no longer operational. Using the input–output framework, HEM calculates the outputs in the entire economy that are necessary for the original final demands. The difference between the original outputs and the HEM outputs (which are smaller than the original outputs) is a measure of the linkages of the deleted industry. "

The methodology is thus designed to provide a metric of the importance of the port, not necessarily a metric for the industry output losses associated with inoperability (e.g. due to a disaster). For the latter, one needs to evaluate how freight can be rerouted to other ports, which if possible, will result in much lower industry output losses compared to the original values provided by the PLOC. However, the latter is not part of this research, but specified as a future recommendation.

Moreover, as the reviewer suggest, it provides an alternative measure of the dependence of a port on industry output. However, the distinction per port is important here, as different ports within a country can have different PLOC measures, as they facilitate goods from different industries and trade can originate from different countries. Hence, some ports are more important/critical for a given country, which is why we believe the PLOC assessment is useful.

We have slightly rephrased the PLOC metric now in line with the reviewer's comment. We have removed any reference to a 'loss in industry output' to it capturing "captures the total industry output and consumption directly or indirectly dependent on the trade flows through a port"

E. Dietzenbacher, B. van Burken, Y. Kondo, Hypothetical extractions from a global perspective. *Econ. Syst. Res.* **31**, 505–519 (2019).

Minor points

-on the logic of the selected commodities: some more explanation may be helpful. Have you considered excluding S2 Fishing? Seems to have specific issues (e.g. fishing by a foreign fleet in national waters of a third country etc).

Fisheries are not excluded, because it is hard to distinguish which part of fisheries are trade and which part originates from a foreign fleet. However, given that the size of trade in this sector is so small, this sector is negligible for most of the results; (1) It hardly contributes to aggregate trade statistics, and therefore for the port-level trade flows and the PLOC and PLIC measures. Hence, we keep it in for consistency, but agree that the fisheries sector is a tricky one when it comes to international trade statistics.

-the abstract can be improved in terms of style

We have rewritten the abstract, which hopefully is clearer now.

-I am certainly not an expert but was wondering if the terms 'downscaling' (line 12 and elsewhere) and 'embeddedness' (line 16 and elsewhere) are used in their conventional meaning?

The term downscaling is often used in this field to scale data from a certain aggregate level (in this case country) to a lower level, either administrative or some other spatial proxy. However, it might be worthwhile to use the term fine-scale representation (L63) to illustrate what we mean and avoided using the word downscaling.

We agree that the term embeddedness is confusing. Although it has been used in similar contexts as we adopt it, embeddedness is also often used in a social economy context, and is more related to the social, political, and religious institutions a certain economy is embedded in. Given that we write for a wider audience, we decided to remove the word embeddedness from the manuscript.

-line 39 ref 8: I am not sure the statement accurately reflects the cited paper.

We have replaced this citation the Walsh et al. paper. This is line with the discussion above concerning the different trade outlooks.

-line 124 the logic of income levels and the ability to afford air transport is not so convincing, this seems to be more related to supply chain characteristics than income levels (see flower exports from Tanzania and Ethiopia).

As the reviewer mentions, the use of air transport is to some extent driven by the type of commodities a country imports or exports. However, alongside the composition of goods, other factors also drive the use of air cargo, such as the reliability of logistics services and infrastructure in place to make air a suitable alternative. To frame it differently, in some cases air transport is not feasible because the trade costs associated with air, and the transit times, can be much higher in low income countries than high income countries.

For instance, evidence has shown that low income countries often do not have the dependent infrastructure in place (e.g. cargo clearance, inspections, warehouses, cooling facilities, logistics services) (see for instance <https://www.worldbank.org/en/topic/transport/publication/air-freight-study>) and lack air freight hubs (from various air cargo integrators such as UPS) and connectivity to fully utilize the benefits of air transport. As a result, it was found that low-income countries pay the highest costs per kilogram of air freight shipped (Vega 2014).

We agree that affordability is not the correct term here. However, the income level of a country does result in higher air freight costs and lower uptake of air freight. Therefore, we have changed the sentence here:

“This difference can be explained by the fact that low-income countries often trade low value bulk goods, for which maritime transport is the only viable option, and relatively few high valued goods that are more often transported by aeroplane” (L90-192)

H. L. Vega, “Air Cargo Services and the Export Flows of Developing Countries” in (2014), pp. 199–234.

-line 137-139: there may not be an overreliance; the logic may go in the opposite direction, limited integration explains the higher share of maritime transport; i.e. it will disappear ‘spontaneously’ once the export flows change.

If we are correct, we think the reviewer refers to lines 126-128 here. After looking at this statement, we agree that reframing this sentence is needed. As the reviewer rightfully outlines, part of the reason that low-income countries have high reliance on maritime transport is because they mainly export low value bulky goods. If their economies diversify, alternative modes of transport might be more suitable/cost-efficient, which could reduce their dependence on maritime transport. Therefore, the overreliance of maritime transport itself might not be the hindering factor. It could still be, however, that the reliability of alternative modes of transport, or just the logistics chain overall, can hinder integration in manufacturing supply-chains. Both absolute transit times and uncertainty in transit times (e.g. delays) are a major risk for just-in-time processes, requiring larger inventories. Therefore, without these logistics services being improved, the relative time differences between modes might not be large enough to use alternative faster modes (land/air) to their full potential. The same holds for changes in the connectivity of different modes (e.g. maritime and air connectivity, construction of highways etc.).

We have rephrased this comments as:

“This difference can be explained by the fact that low-income countries often trade low value bulk goods, for which maritime transport is the only viable option, and relatively few high valued goods that are more often transported by aeroplane⁴⁵. Even within the same continent, such as in Africa, maritime transport is often the only feasible mode of transport as the road infrastructure lacks the reliability and capacity for efficient trucking, and passing of border crossings are often time consuming^{46,47}. Therefore, the integration of low income countries into complex manufacturing supply-chains, which critically depend on just-in-time logistics services⁴⁸, could be hindered by their overreliance on maritime transport, which is considerably slower than air transport^{49,50}.” (L190-198)

-line 222: ignores the possibility of transit cargoes

This is now being added in the method described in the new manuscript.

-there are some inevitable simplifications such as the capital city and the maritime distances. Would be good to understand their impact on the findings. For some countries, the impact seems relevant (e.g. Russia, Canada, US, Colombia, Mexico).

Line 581: perhaps explain in more detail the underlying logic?

The objective of the modal split model is to estimate the share of maritime transport in every bilateral trade flow based on the cost differential between modes (associated with the time and distance it takes

to transport goods) and other factors (e.g. income, countries being neighbours etc.). Therefore, for every country-pair a generalized cost should be established. Instead of using the capital cities, we have now used the nightlight weighted centroid of countries. For the maritime distance, we have now included the actual distance it takes to transport goods between countries, including potential transshipments if no direct connection exist.

Both additions improve the modal split model with better validation statistics and external validation than before.

For the maritime transport model, we go into more detail and look at multiple subnational centroids per country. This to take into consideration that for some countries, part of the flow might go via a port in another country if it has a better/cheaper connection to that centroid (such as the ports of Antwerp/Rotterdam for Germany).

F1: the bar chart in the figure is a bit hard to follow

The bar chart reflects the distribution of countries with respect to their percentage of maritime imports and exports. However, we think the bar chart actually does not provide much extra information to the plot since we have a geographical plot and a boxplot that captures the same information. We therefore decided to remove this bar chart.

Reviewer #2 (Remarks to the Author):

We would like to express our gratitude to reviewer 2 for this extensive and extremely helpful review of our paper. The reviewer pointed out some discussion points related to the framing of the paper, how our research fits within the existing literature, and the methodological choices and assumptions made throughout the paper.

Please refer below for our detailed response, in which we have tried our best to address all the comments made by the reviewer.

The manuscript under review reports on an ambitious project to link trade flows to specific ports. This is a welcome addition to our knowledge given that much of the trade literature seems to ignore that trade in goods 'takes place' and materializes in the form of traffic that makes use of infrastructure. Moreover, it adds to the literature on ports and maritime transport, where the link with economic activity in the hinterlands of ports is usually discussed with less detail. It is promising that the approach goes beyond case studies, which are also abundant in the field of Global Value Chains, and aims to cover the entire world and all relevant industry sectors. The core of the work is based on a multi-region input-output table, the Eora MRIO. A first step is to remove non-maritime trade flows by estimating the share of each flow that is transported by ship, air and road, using a modal split model. Subsequently, the maritime country-to-county trade flows are attributed to port pairs on the basis of a detailed database of actual vessel traffic. The resulting database contains disaggregated data on 157 countries, 1132 ports and 11 industry sectors. Note that a variety of data sources is used throughout the paper, not just the MRIO table. Finally, the data is applied to estimate (1) the share of maritime transport in global trade, (2) the share of trade (disaggregated per sector) that passes through specific ports, (3) the degree of specialisation and the coreness of ports, (4) the port-level output coefficient (PLOC), 'that captures the total industry output directly or indirectly dependent on the trade flows through a port', and (5) the port-level import coefficient (PLIC) which gives 'the direct and indirect (through interindustry dependencies) imports per port needed to produce the domestic consumption and (re-)exports in an economy'. My comments relate to (1) the aims of the paper, (2) the conceptual framework, and (3) empirical challenges.

Aims of the paper

The abstract indicates that the aim of the manuscript is twofold. On the one hand, it offers ‘a comprehensive methodology to downscale global trade flows to a port-level, which allows us to incorporate all significant ports in a global supply-chain database’, on the other hand, the aim is to evaluate the criticality of ports for the economy, and more specifically, to obtain a ‘better understanding [of] the feedback and dependencies between ports and the economy, thereby supporting long-term resilient infrastructure planning’.

The first aim is to develop a methodology to obtain a new type of database. Regarding this aim, surprisingly little is said about existing databases. For quite some time, statistical agencies, consultancies as well as other companies provide data on trade flows, port throughput, and the economic structure of countries and regions. As a consequence, the obvious question is, why has no-one come up with this or a similar dataset before? One reason might be that many sources are behind paywalls. The current data can be presented as a freely available alternative to products from commercial players such as Clarksons, Containerisation International, IHS Global Insight or Seabury. Another possible reason is that we had to wait until the development of global supply chain databases such as the Eora MRIO. I leave aside the possibility that the reason for the current absence of such a database is its irrelevance. In other words, the article needs a clear position relative to the state-of-the-art. Reference is made to studies in different relevant fields including Global Value Chains, port studies and MRIO, but what is lacking is an explanation why the work reported here is presented today, and was not done before. My guess is that there is more at stake than the empirical challenge to link all the different data sources.

We thank the reviewer for bringing up this comment. Although we should have reflected on this more explicitly, we did not think it was appropriate to have a comparison of datasets from commercial providers for a high-level article. Having said this, we do think it is critical to understand what is currently out there and why such a database could not have been created before.

In short we can distinguish the follow relevant datasets:

- MRIO: Global multi-regional input-output tables in value terms. Although it does have a global coverage and sector disaggregation, the mode of transport used is not reported in such database.
- Global trade data: UN Comtrade recent trade data covering all modes. Only recently (last year) mode of transport data is available in UN Comtrade for those countries that report this (~50 countries). On top of UN Comtrade data, BACI harmonized trade data exist which have balanced imports and exports and corrected from reporting errors. Both data sources are on a country-level with high sector disaggregation.
- Bill of Lading data: Detailed Customs data that includes shipment level information (mode of transport, port of entry, sector, sender, receiver, etc.). Such data can be purchased from for instance IHS Markit, but is only available for 14 countries.
- Monthly customs data per port: Detailed monthly customs statistics on imports and exports per port and sector. Only a few countries do report this data (e.g. US, UK, Japan, New Zealand, Brazil).
- Port-level aggregate data: Some commercial data products exist, such as the Drewry’s ‘Port and Terminal Insights’, Lloyds top 100 container ports, IHS Markit’s Ports and Terminal Guide. Although this detailed information is helpful, traffic/volume information is often only available for a certain number of ports/terminals (e.g. Drewry is 250 ports and terminals), covers only the container market and therefore reported in TEU only, or does not provide a further sector disaggregation. Therefore, while this is useful for validation (although prohibitively expensive), it does not provide a global coverage of all ports and sectors.

In short, we therefore use a mix of datasets to come up with a structured way to fill in the data gaps and create a consistent new global dataset that as the overarching objective to be compatible with the MRIO. To do this, we do:

- 1) Use mode of transport from UN Comtrade to fit the modal split model. Although mode of transport data is freely available, downloading data for multiple years is hard, especially if a large sector disaggregation is needed (due to limitations on number of rows you can download). Because of our collaboration with the UNSD, we were able to obtain all mode of transport data from the server. We combine this modal split model with the BACI database to have a harmonized maritime mode of transport database.
- 2) The missing link to further disaggregate the maritime trade flows to a port-level is the Automatic Identification System (AIS) data and the maritime transport model we developed. This allows us to actually provide (estimated) port-level trade data for the different industries. We use the Customs data to validate this approach. The use of AIS data in (maritime) economics is relatively new, and we have been the first developing a global trade prediction dataset based on this (at least the first published one).
- 3) Most importantly, we have now developed a global maritime transport model which predicts the allocation of maritime trade flows on the maritime network.
- 4) Altogether, this all allowed us to (i) make sure data is consistent with official trade data, (ii) present a generalized approach with global coverage in terms of ports and have the correct sector classification to couple the data to the IO tables.

Therefore, there were significant technical challenges to do as well as major shortcomings in external data sources to do such an undertaking.

In order to better communicate this, we reframed the introduction to include part of this discussion. We still do not think that a detailed comparison (like described above) fits well within the main manuscript given the journal's aim and formatting. If the reviewer believes it is needed, we could write a Supplementary Note on this.

The manuscript under review is based on a paradigm that is not mainstream in (trans)port economics, to say the least. Port economics is almost a synonym for neoclassical port economics and the promotion of marginal cost pricing, and the field has a legacy of fighting protectionist tendencies of governments that want to promote their own, national ports.¹ It is known that neoclassical economists often see input-output analysis as an instrument of the central economic planning they oppose, while input-output economists argue that neoclassical economists ignore the structure of the economy.² Seen from that perspective, the aim of the paper is not just to introduce a dataset or to estimate the impact of disruptions, but to import a different paradigm. Note that examples of input-output analysis can be found in the literature on ports, but mainly to investigate the port sector as an industry.^{3–5}

The reviewer makes an interesting point here. We agree that the field of economics, or rather the various subdomains within economics, have certain paradigms that are dominant. And that it can sometimes be difficult to introduce a different way of thinking within a certain subdomain. Perhaps we distinguish ourselves here as being opportunistic economists instead of neoclassical or input-output economists. We simply took, to our knowledge, the best available data and methods to find an answer to our research problem. We do, however, think it is beyond the scope of the paper to talk about the introduction of a new way of thinking in port economics. We feel that would require a full-blown literature review and historical assessment as, for instance, done in Vanoutrive and Zuidhof (2019). We believe this would dilute our key outcomes too much. We do hope, of course, that our study will be a starting point for many new economic port studies. We have tried to align our research as much as possible with the framing as for instance discussed in Robinson (2002), although more focused on the supply-chain than the logistics chain.

Vanoutrive, T. and Zuidhof, P. W. (2019) 'Port Economics in Search of an Audience: The Public Life of Marginal Social Cost Pricing for Ports, 1970-2000', *Oeconomia*, (9–2), pp. 371–394. doi: 10.4000/oeconomia.5822.

Robinson, R. (2002) 'Ports as elements in value-driven chain systems: The new paradigm', *Maritime Policy and Management*, 29(3), pp. 241–255. doi: 10.1080/03088830210132623.

The second aim, as the title suggests, is to provide insights in the criticality of ports in international trade and global supply-chains. If I'm correct, 'criticality' is used as a synonym for resilience,⁶ however, 'Most analysts use resilience in a common sense and non-rigorous fashion'.⁷ Assuming that the five aforementioned applications reveal what the authors mean by criticality or resilience (and port embeddedness), the definition of the concept remains vague when looking at these five interpretations.

We would like to stress that criticality and resilience are not the same concepts here and we have purposely avoided to use the term resilience in this research. Criticality here is related to the importance of a given port/maritime transport for a country through its dependencies on this port/maritime transport. (Port) resilience, on the other hand, has to do with the ability of ports to cope and recover from adverse shocks (e.g. a natural disaster), which is related to, among others, the engineering standards, policies and protocols in place to respond and recover, available budgets for reconstruction/reopening ports, rebuilding times and lead times of components in case of physical damages. Moreover, from the trade perspective, it has to do with the ability to reroute goods, ports to recapture part of their disrupted trade, and so on. Although resilience has been used in other contexts as well, and as the reviewer mentioned often in a non-rigorous fashion, we would like to emphasize that these are different concepts that should not be used interchangeably.

We agree that the way criticality is framed now and the link to the example applications is not clear. First, we have decided to remove the term port embeddedness in line with the Reviewer 1's commentary. Now, we have tried to more explicitly state what we mean with criticality in terms of the various dimensions that economies dependent on ports for their well-functioning.

(1) the share of maritime transport in global trade seems to be used as a crude way to stress the importance of maritime trade, reminiscent of the references made to UNCTAD's Reviews of Maritime Transport:

'For an economic region such as the European Union, shipping accounts for 80% of total exports and imports by volume, and some 50% by value.'⁸

'Maritime shipping is the backbone of world trade; it is estimated that some 80 percent of all goods are carried by sea. In terms of value, global maritime container trade is estimated to account for around 60 percent of all seaborne trade, which was valued at around 14 trillion U.S. dollars in 2019.'⁹

'According to the United Nations Conference on Trade and Development (UNCTAD), more than 80% of global trade by volume, and more than 70% by value, travels by ship. UNCTAD put the volume of seaborne trade in 2015 at 10.3 billion tons, with a combined value of \$25.3 trillion.'¹⁰

Given the rigorous and academic nature of the work, I expected a note on how the methodology differs from that of, in the first place, UNCTAD, on whether the finding that '55% of global trade (by value) is maritime' corroborates existing estimations (as a kind of triangulation), and on the meaning of these figures (in what sense do these measure 'criticality'?).

There are two points to unpack here.

First, the UNCTAD related publication makes the case that the importance of maritime transport for a country is related to the percentage of maritime trade used in trade. We want to make the case that the importance or criticality of maritime trade goes beyond such simple aggregate metrics as it ignores how this maritime trade is produced and used in the economy by supply-chains. A dollar value of maritime trade of a particular sector in a particular country may be used differently than a dollar value of maritime trade of another sector in another country. However, this cannot be captured using these traditional metrics. Therefore, we think the explicit link between maritime trade and the economy provides a more comprehensive picture of the criticality of ports.

Second, the commonly accepted number of 60-70% of global trade in terms of value being maritime is actually a controversial number. For instance, UNCTAD uses this number in their annual reviews without actually deriving it themselves. According to their website, this number, and maritime trade statistics in general, is on the “*basis of data supplied by reporting countries, as published on the relevant government and port industry websites and by specialist sources.*” We have been in touch with the UNCTAD people that constructed the maritime trade data and they mentioned that there are several issues with this. First, they use data from port authorities, Clarksons Research, national statistics officers, ministries of annual transport reports, etc. In a lot of cases these data sources have to be corrected or adjusted. For instance, country data often does not include oil trade. Second, domestic and transshipment traffic is sometimes included in their data, which should be filtered out to avoid double counting.

To add to this, if you reverse the equation (so not knowing maritime trade in value), the 70% does not make sense at all. For instance, IATA states that 35% of global trade in value is by air, and you combine that with the fact that around 10-15% is by means of land, maritime trade cannot be more than 50-55%. Therefore, the number of 50% we end up with now is much more in line with expectations than the 70% presented by UNCTAD. In terms of tonnes-km, 70% would probably make much more sense.

We agree that both a reflection on this number is needed. In short, (1) reliance on maritime transport is only part of the importance of ports, and (2) even deriving maritime trade data is controversial as consistent information does not exist globally. The latter is also confirmed by UNCTAD as I quote from their email to us: “*All this to say, there is no centralized database for maritime trade data, and you have to collect these across different sources and make many assumptions to adjust these figures.*”

However, we do not think this journal is a suitable outlet to discredit the numbers provided by UNCTAD. We do, however, now compare our results with results presented by UNCTAD to show that both are in the same ballpark. Again, we are happy to add a brief Supplementary Note on this if deemed necessary.

Similarly, the fact that low-income countries have a lower share of aviation, export relatively more mining products, often depend on the export of a limited number of commodities, and are less well integrated in global shipping networks (see also (3)) is an established fact. This does not reduce the relevance of the observation, but requires that it is presented as corroboration of existing information.

We agree that this is not new information, but merely to explain our results to wider audience.

Finally, the conclusion that 'On average, every 1000 dollar increase in final demand of an economy leads to an additional 2.2 dollar of imports going through a given port' and that 'On a country-level, for every 1000 dollar increase in final demand, ports (i.e. all ports in a country) will experience a mean (maximum) 18.3 (108.3) dollar increase in maritime imports.', might also be framed as a critique of claims that maritime trade is crucial (2.2/1000 does not sound impressive). Some additional interpretation might be required.

We agree. First of all, we have revisited the PLIC formulation, resulting in slightly higher values. Although we think the numbers on a port level are relevant, the main relevant fact is the import coefficient on a country-level, as it captures how maritime imports are needed to produce changes in final demand. Therefore, we have rewritten this section (L412 – 432), which is focused on the country-average values that are higher than previously reported.

(2) the second measure is the share of trade that passes through specific ports (e.g. ‘9.3% of the total industry output depends upon trade flowing through only the top 10 ports’). Here the focus is on ports, and a ranking of ports is given.

We have now removed this statement. This because transshipment are now included in the model, so simply adding up the numbers per port like we did before is not feasible anymore as it would double count a lot of the industry output.

(3) also the third interpretation of criticality takes a port's view and is measured as the degree of specialisation and the coreness of a port. This interpretation follows the logic of network analysis (and graph theory). Here too, reference can be made to existing measures (e.g. UNCTAD's liner shipping connectivity index).

Based on comments provided by reviewer 1 and after revisiting this analysis, we have decided to remove the specialisation/coreness from the manuscript.

(4) the port-level output coefficient (PLOC), is based on input-output analysis,

(5) the port-level import coefficient (PLIC), is like the PLOC, based on I-O analysis.

Although the two last coefficients are measured at the port-level, the emphasis is on the country-level (the 'economy', and are closely related to country-level import coefficients (CLIC)). Hence, measures of 'criticality/resilience' are found for different things, which leads to the recommendation to be clear of what the 'criticality' is estimated.

We admit the confusion. What we mean by criticality is mainly the reliance on maritime trade and the link between the maritime trade on a port level and the supply-chains. Both say something about the importance for the domestic/global economy, though in different ways, which we think is more encompassing than just saying how much trade is imported/exported or goes through a port (port throughput). We have now emphasized that we try and 'capture the different dimensions of the criticality of ports for domestic and global economies that are not captured in aggregate port-level trade statistics.' (L86-87) and (L439-442).

While the port numbers show the importance of individual ports, the country aggregated values help in explaining differences between countries in the way their supply-chains dependent on port/maritime trade. We think that including both is essential here.

A third (potential) aim, contributing to environmental research and policy, is only briefly mentioned (e.g. the data can help to 'connect environmental footprints with commodity flows', 'estimating the carbon emissions embedded into maritime transportation' and 'maritime decarbonisation strategies'). Nevertheless, besides the affiliation of the authors and the scope of the journal ('research from all areas of the natural sciences'), the MRIO-framework fits well with environmental aims.¹¹ Surprisingly, no critical remarks with regard to environmental impacts were made about the observation that 'Future demand scenarios project a near tripling of maritime trade volume by 2050' (besides, the first paragraph of the paper looks as if it was copied from promotional material of a port, in contrast to the rest of the article). However, taking the environmental road might imply different methodological choices, like focusing more on weight than on value.

We mentioned the environmental application here as vaguely similar methodologies have been proposed for trying to allocate maritime emissions, as a result of maritime transport between countries, to a given economic sector. The best examples here are a study that looks at China-USA maritime trade (Liu et al. 2019), and one looking at the Brazilian exports (Schim van der Loeff et al. 2018). For these two studies to relate the maritime transport emissions to economic sector, they had to make use of detailed bill-of-lading data (see discussion above), which is therefore not transferable to other countries as this data is not available. Hence, our methodology could therefore bridge this gap and provide a generalisable approach to extend this approach to other countries. However, in order to calculate emissions, a detailed methodology has to be followed that estimated the carbon emissions embedded in trade for different types of vessels (in terms of size, age, fuel, engine type) between all country pairs. The latter is specialised work and although could certainly be an extension of our work,

is not something that can be easily added without deep domain knowledge. Therefore, leaving it as a future recommendation only is justified in our view, and we hope that other researchers will take up our model output to work on this.

H. Liu, *et al.*, Emissions and health impacts from global shipping embodied in US–China bilateral trade. *Nat. Sustain.* **2**, 1027–1033 (2019).

W. Schim van der Loeff, J. Godar, V. Prakash, A spatially explicit data-driven approach to calculating commodity-specific shipping emissions per vessel. *J. Clean. Prod.* **205**, 895–908 (2018).

And what about the concept of international trade itself? From a transport and environment perspective, I'm interested in ton-kilometres, vehicle technology and modal split, not in whether goods cross national borders. I recommend to mention whether environmental impacts fall outside the scope of the paper, or alternatively, can easily be included in the methodology.

We agree that we should have been clearer that environmental impacts are not included in this work. Although we have now included numbers of ton-kilometres and the use of hinterland transport for to connect demand centres to ports, we leave it up to other work to further investigate the environmental implications of this.

“Although beyond the scope of this work, this measure could help evaluate the potential losses within supply-chains networks if ports are disrupted by a shock, and help quantify the emissions associated with transporting goods towards and from specific ports are linking to foreign or domestic supply-chains.” (L380 – 384)

Conceptual framework

An interesting feature of the manuscript is that it combines insights, concepts and data from different fields, including MRIO analysis, port and maritime studies, and the global value chains approach.

However, this might come at the cost of conceptual clarity.

Many concepts are rather clearly defined in the article (e.g. (dominant) transport mode, PLOC, PLIC,...). As was discussed earlier, some confusion might rise regarding the exact definition of criticality and resilience (and whether this is the same as port embeddedness).

We tried to clarify this now, also in line with the suggestions made above. In particular we have rewritten the introduction and problem statement and wrote an introductory sentence to every result paragraph so reader can better follow the overall narrative. We have that this has brought more clarity to the structure and framing of the research.

Taking into account the background of the target audience of the journal, it might be advisable to make clear what the ‘economic’ concepts exactly mean. I recommend to make clear how the ‘world’ is approached, i.e. using the MRIO framework. If someone applies a gravity model, the idea behind the model is pretty easy to explain, i.e. the amount of interaction (trade) between two regions depends on the size of the economies and the distance. Something similar can be done for MRIO. A picture (flowchart) can be used to explain the essence of the chosen framework, which needs to be accompanied by an argumentation why this view on the world is the most appropriate. Discussions of ‘Trade in Value Added’ illustrate that even the concept of ‘value’ might be measured and interpreted in various ways.¹² A simple diagram might help readers to understand what concepts mean and how these relate to each other, think of (total) industry output, intermediate/final demand, direct/indirect imports, exports, direct/final/domestic consumption, forward/backward linkages, feedback,...

Agree, we have described this more clearly throughout the manuscript. We have also made a simplified flowchart which we included in Supplementary Figure 1 for reference.

Specifically, the concepts of dependency (and influence) need careful introduction as these can be interpreted rather mechanically, and can find easily their way to newspapers and other media (for example, 'In total, 41.9% of global industry output depends on the trade flows going through port').

To avoid confusion, and because of the issue around double counting now that transshipments are added, we have remove this number. But in general, we have tried to be clearer with our definitions.

Another relevant concept is 'losses'. When it comes to the use of the results for policy, the article aims to contribute to a better understanding of 'how demand shocks (e.g. pandemic) ripple through the economy' and of 'potential losses within supply-chains networks if ports are disrupted by a trade shock (e.g. natural disaster or labour strike).' To this end the paper mobilises 'the port-level output coefficient (PLOC) that captures to what extent industry output is reduced (both domestically and globally) if a particular port is hypothetically extracted from the I-O table' (short-term), and 'the port-level import coefficient (PLIC) that measures the increase or decrease in port-level trade flows when demand changes in an economy' (longer-term). Moreover, 'by coupling this framework to a disaster impact model, the economic-wide losses (domestic and global) from port disruptions could be assessed, including the future losses due to climate change (e.g. sea-level rise, increased frequency of flooding)'. The approach could also 'help refine the future investment needs of port infrastructure and evaluate the changing dependencies between ports and the economy' (long-term infrastructure planning). To have an idea of the short-term impact of disruption, I-O based approaches can give an indication of the industries that will be affected by the disruption, together with an order of magnitude. As has been discussed in the literature, in the short term, substitution of one input by another is less likely, just like major reorganizations in production. When disruptions last longer, the structure of I-O relations will probably change, which makes it less meaningful to hypothetically extract a port from the I-O table.⁶ For the longer term (and not just for reductions in industry output, but also increases), it is hypothesized that an increase in demand has an impact on port throughput, without taking into account that changes in ports (that affect the generalized transport cost) have a fundamental impact on the structure of the economy. The latter is very common, and decreasing transport costs (associated with investments in infrastructures such as ports) are seen as the driver of globalization and as a force that changes the structure of economies and the spatial pattern of economic activity (among others in the New Economic Geography literature^{13–15}). From a neoclassical perspective, the analysis is presumably seen as too static, and too mechanical, to say something meaningful about the effects in the longer term. But this is a matter of choice of paradigm, and as long as someone is open about the adopted paradigm, and aware of the shortcomings and criticisms, choice of paradigm is not in itself a basis to reject a particular approach. What can be criticized is that the current version of the manuscript is not so clear about the importance of paradigmatic differences. From a port economics perspective, this is not simply an empirical exercise which leads to a new dataset, but a different paradigm that is applied (see discussion above). Turning back to the policy-oriented applications presented in the manuscript, a key concept is 'economic losses' which are caused by disruptions. The idea of a loss presupposes a reference base, usually the situation just before the disruption appeared, and a counterfactual, i.e. what would happen if the disruptive event would not take place (business as usual). Natural disasters (including floodings caused by climate change) and labour strikes serve as examples of disruptions.

We tried to make the link with our work and its relevance for disaster risk research, as it is the one of the clearest applications of our output, but we have consciously tried to avoid the term losses throughout the paper for several reasons. Foremost, we avoid losses because the current research output can help inform such risk analysis, but additional data is needed to quantify the losses. For instance, as we set out in the discussion about the term 'resilience', in order to talk about losses additional information and modelling steps are needed in this paper. So, even though, this type of data is the first building block for modelling such supply-chain losses through ripple-effects, we only

present it as one possible outlook for application. This to say, we tried to keep the framing and policy relevance quite broad, as the dataset has multiple applications, in order not to create the impression that this dataset has been developed for a single application. However, based on the comments of the reviewer, we feel like we have not yet successfully conveyed this. Therefore, we have rewritten the introduction to better explain this.

However, what about the corresponding counterfactuals (the *ceteris paribus* question)? Someone who builds a port in a hurricane-prone region should know that there will be a certain number of days that port operations will be 'disturbed' by bad weather. It's hard to predict the exact number of hurricane days in a given year, but the counterfactual of a port in a hurricane-prone region without hurricanes is a contradiction in terms. When 'natural' disasters are caused by human economic activities, as is the case with climate change, then the counterfactual of an identical economic structure (I-O table) without greenhouse gas emissions is also not very likely. The same can be said about labour strikes. In a perfect neoclassical market, wages reflect the value of a person's labour output. When wages are lower than this value, this might induce a strike. Low wages in ports are beneficial for the industries that depend on them since transport costs are lower. The counterfactual of low wages without any action from the labour side is, again, unlikely. We can continue the long-lasting debate on whether monopolies on dock labour distort markets, or whether it's more of a monopsony, but the point is that many strikes are not something external to an economy. I got the impression that the concept of economic loss is linked to external disruptions. What needs to be explained, then, is what a disruption and the resulting economic loss actually are.

This is a very good point and also reflect the long-lasting debate in the disaster risk field on creating good counterfactual scenarios for disaster analysis. However, linking it back to our discussion above, we do not want to explicitly 'model' losses in this analysis, and hence also stay away from the discussion on counterfactuals for purpose of this paper.

Empirical choices and issues

The methodology in the manuscript is presented as a 'spatially explicit analysis', and from an international trade analysis perspective, this is a step forward. However, the definition of 'economies' is rather traditional as 'economies' are equated with national states. Paul Krugman once asked the question 'Why is the process that puts a bottle of French wine on a Berlin table very different from that which puts California wine on New York tables?' (p.49).¹⁶ In this respect, following remark in the manuscript is relevant: 'European countries have a much lower share of maritime transport, mainly due to the large flows between European countries'. The spatial aggregation of economic data might have a major impact on some of the results, as has been discussed elsewhere for MRIO analysis.¹⁷ Especially for large countries such as the US and China this might matter, as it does for integrated trade blocks and custom unions such as the EU. The use of capital cities as the location of airports might also cause some bias (cf. 'the geographical distance between capital cities'; 'the fact that we use capital cities as centroids, thereby estimating higher transport costs for road transport').

Please see below for the discussion on the spatial units. In terms of the use of capital cities, this might induce some error but we expect this to be minimal. Based on the reviewer's comment, and a comment of Reviewer 1, for the modal split model we have replaced the capital city with the economic activity-weighted centroid of a country (based on available country centroids from nightlight data). For the transport model, we have decided to use subnational administrative regions as suitable centroids to better capture how different hinterlands use different ports.

Also the inclusion of a large amount of small island states without weighing factors can bias some findings. For some, this is not only a technical issue related to the division of the world in arbitrary spatial units (~ Modifiable Areal Unit Problem),¹⁸ but also a substantive issue as 'national economies' do not exist according to among others World City scholars who found inspiration in the work of Jane

Jacobs, and developed a framework to use cities as unit of analysis.^{19,20} I suggest to note the problem with spatial aggregation somewhere in the text, and to check what the impact might be.

We thank the reviewer for pointing this out. We agree that in many applications, including ours, national economies are not always the best spatial units for analysis, and subcounty administrative boundaries, or cities are better spatial subdivisions. However, we are forced to work with national economies, given that the available trade (absolute trade and mode of transport) and IO tables are on a country level.

We do agree that this ‘issue’ deserves special attention and we have therefore mentioned it in the Methods section. We do however note that a sensitivity analysis would be difficult here given the input data we have.

“Throughout the analysis, we use national economies as the spatial-level of aggregation for most of the analysis, which we further disaggregate to the port-level. This because the international trade data and global supply-chain database are constructed on a country by country basis, restricting using subnational economic data. We do recognise that this might bias some of the results as some interpretations of the results might be related to the size of the economy.” (L548 – 553)

The same holds for ports, which are seen as nodes in value chains.²¹ But here too, the delimitation of port areas is not without its challenges. The territory of some port areas is located in two countries (Copenhagen-Malmö; North Sea Port), or includes locations separated by a large distance. If the French government would decide to make of the Port of Le Havre-Antifer a separate port, this would be a highly specialized port (oil). Hence the question, to what extent does the ‘arbitrary’ aggregation of terminals in ports affects the results? Some even argue that it is rather the terminal than the port that has become the relevant unit of analysis.²² Alternatively, one can aggregate data in port ranges (e.g. the Hamburg-Le Havre range),²³ and ‘port regionalization’ has been widely debated in the ports literature.²⁴ I recommend to add a note on the definition of ‘port’ applied in the paper, and a reflection on the possible effects of other ways to aggregate data.

This discussion point relates to the discussion on the country-level data, so we have decided to add a sentence on this as well. We have tried to be as consistent as possible and used the spatial delineation of ports as used in the World Port Index. Aggregation in ports ranges and ports expanding multiple areas are difficult given the connection to the national IO tables. But we still agree that the definition of port is non-arbitrary and should be defined.

“Further, throughout this research, ports are defined as one or multiple terminals within a specified port boundary, which have been delineated in line with the World Port Index, the most widely used database of ports.” (L536-538)

In some cases averages are taken (e.g. ‘We find a coreness value per port per sector and construct the total coreness value of the port by taking the mean of the coreness values across sectors.’). I’m rather skeptical towards taking the mean of values across sectors, given the difference in importance of sectors, and the uneven spatial distribution of sectors. Are alternative measures taken into consideration, or is some kind of sensitivity check carried out?

Based on a comment provided by Reviewer 1 and a comment made below about the coreness proxy, we have decided to remove the coreness metric from the analysis.

Small comments

-‘Overall, specialised ports have lower PLIC (Spearman’s rank correlation: -0.24), while the coreness

positively scales with the PLIC (Spearman's rank correlation: -0.50).': Is it correct that both correlation coefficients have a negative sign?

The second relationship is indeed positive. Thank you for pointing out. However, this is now removed from the analysis since the specialisation/coreness has been removed.

-For instance, Hurricane Katrina (2005) and the Hanshin-Awaji earthquake (1995), damaging the ports of New Orleans and Kobe, respectively, resulted in large shocks to maritime transport networks and global supply-chains. Moreover, the COVID-19 pandemic, disrupting supply-chains globally, reduced global maritime transport by 7.0-9.6% during the first eight months of 2020.': the recent grounding of the Ever Given in the Suez canal can be added as an example.

We now included these more recent examples, thank you for suggesting.

- 'whereas Middle-Eastern and South American rely more': add 'countries' between 'American' and 'rely'

This has been changed now.

- 'The large unevenness in terms of trade flows suggest a hierarchical structure of trade within the port-to-port trade network.': is 'hierarchical' here used as a synonym for 'hub and spoke', which also implies that the type of vessels differs between ports.

After reading this sentence again, we realize that we misused the term hierarchical here, as hierarchical often means that different networks work independent from each other (on different layers). We do mean something more closely related to a hub and spoke system, although not directly, as hub and spoke networks in the maritime network literature encompass the system in which large vessels distribute trade to large transport hubs, after which smaller feeder vessels further distribute this trade to the small ports that cannot accommodate these large vessels. Here, we try to emphasize that a small fraction of ports drive the vast majority of sector-specific trade, illustrating that economies of scale are clear and that there is a hierarchy of ports in terms of how they contribute to global sector-specific trade. However, this is also different between sectors.

We have rewritten this as:

"These sectoral heterogeneities do not only reflect the differences in the clustering of industries, but also economies of scale present in the transport of some goods^{54,55}. For example, while for some highly concentrated sectors the vast majority of goods will be transported between a subset of core ports, other less concentrated sectors will use a more decentralized transport network." L301-305

- 'The products that are imported through a port are directly consumed in a country or are used in production processes to produce goods for domestic consumption or export.': is it possible that a product is imported and exported without being 'used in production processes'?

Pure transshipment of products (or re-exports) happens, but it not properly accounted for in IO tables (Lankhuizen and Thissen, 2019), so we prefer not to mention it here.

Lankhuizen, M. and Thissen, M. (2019) 'The implications of re-exports for gravity equation estimation, NAFTA and Brexit', *Spatial Economic Analysis*, 14(4), pp. 384–403. doi: 10.1080/17421772.2019.1623419.

- 'the mode-specific transport costs per tonnes per km (US\$/t/km)': to what extent is taken into account that these values differ between countries (cf. road speeds), and especially for maritime trade, between vessels of different sizes?

In our new maritime transport model, several improvements with respect to the costs have been made.

First, we now have country-specific hinterland transport costs based on a database obtained from the ITF-OECD.

Second, we have distinguished maritime transport in five vessel types that all have different speeds and costs. Moreover, ports have specific cost components (e.g. handling costs, dwell/turn around time). Unfortunately information on the cost w.r.t. economies of scale is not available (e.g. transport costs between specific port pairs).

Altogether, we believe the cost representation is considerably better than in the earlier version. We kindly refer to the new Supplementary Information here.

-Modal choice model: if the goal is explanation and interpretation, the choice of variables is important. However, when the goal is to find a function that fits the data well, adding more variables (and multicollinearity) is not an issue. One can add, for example, a dummy variable for each origin and destination country, instead of using variables such as GDP per capita. When the modal choice model is simply a step towards a database, more variables can be added to increase model fit. Furthermore, dummy variables that represent country pairs or considering countries in the same trade block or customs union as neighbours (instead of 'neighbouring country dummies') can be considered.

Unfortunately, the data we have to fit the model only includes a subsample of all possible country pairs and hence we needed to pick enough variables to capture the heterogeneity between country pairs, but not including variables that cannot be extrapolated to the full trade dataset. In a more data rich environments, including the country pair dummy would result in considerable improvement, but in our model we could not do this because of our sample only includes a subsample of all country pairs. That is why GDP/capita was a good intermediate solution. We also tried adding continent pair dummies, but we found that this did not improve the model fit enough to include it in the final model we adopt. The neighbouring country dummy here is to include that countries that are neighbours will very likely use road transport despite the distance between capital cities. Being in the same customs union will not very much affect the likelihood of taking one mode over the other, so we have excluded this from the analysis. However, we do now include border time to the cost function, which therefore partly captures this already.

-'For landlocked countries (which are not included in the port-to-port dataset), we use the road network to connect landlocked countries to the closest 15 ports in 15 countries, to allow landlocked countries to use ports in different countries (e.g. Switzerland can use the an Italian port for trade from Asia and a port in Germany for trade from Scandinavia).': If I'm correct, this procedure is only followed for landlocked countries. However, it is not ridiculous to claim that Rotterdam and Antwerp are 'German' ports, although these are located in the Netherlands and Belgium respectively. Especially Germany's Rhine-Ruhr region has good connections to ports in the Low Countries.

This is now being included in the revised version of the methodology using the maritime transport model.

-'We validate how the well the disaggregation method': delete 'the'

Thank you, this is corrected now

-'although the methodology generally concentrates to much trade in smaller ports.': 'to' -> 'too'?

Thank you, this is corrected now

-Trade flows included in the MRIO table are not always similar as those included in the BACI trade database: here too, some methodological interpretation and discussion might be relevant. HS is developed by the World Customs Organization (WCO), and is based on the nature of the commodity which fits well with its use by Customs. However, for economic analyses, the logic of customs is not always appropriate, and for example in SITC the end use of a good defines its classification in a category. This illustrates how conceptual and measurement issues relate to each other. Another reported issue is that customs-based import data are in general more reliable than export data since customs in the first place check goods that enter the country.²⁵ Furthermore, trade data stems from countries and different data collection practices exist in different countries.²⁶

As the reviewer points out, this is a general limitation of using trade data, in particular UN Comtrade data. This is the reason why we have used BACI trade data as it is an harmonized trade database that already includes data corrections to the original data (for instance to balance imports and exports, fix reporting errors etc.). On top of that, BACI and MRIO differ because the construction of a global MRIO involves a data reconciliation process. In this process, MRIO data from multiple sources are combined and a balanced (well matched) overarching MRIO is created (often done use RAS-type methods). This creates trade flows that do not exist in the trade database since they are there solely because of the balancing of data. However, these trade flows are often quite small.

-Although the work done is impressive, I suggest adding a warning for those who want to use data from a particular flow of goods through a particular port. When looking at the information in the supplementary files, the difference between estimated flows and actual flows can be substantial for particular cases (see ‘Supplementary Fig. 9 | Validation trade allocation algorithm’, ‘Supplementary Table 6 | Validation of the derived maritime trade shares for European Union countries’).

We agree, we not claim to have a perfect database, and the main focus is to develop an approach that results in globally consistent estimates, understanding dependencies in the maritime transport network, and the opportunity to link it to the IO tables. The above comes at the expense that errors accumulate, which can vary across ports and countries. We will add a disclaimer to the Mendeley Data product we intend to publish alongside the paper to explicitly state what the intended purposes are for the data. Indeed, the dataset is more suitable for broad-scale global analysis than port-specific analysis.

Conclusion

The work done is a major effort to develop a database that links port throughput to economic activity. Such a database can have many applications. In general, I recommend to:

-be more explicit about the chosen paradigm, its relation to alternative conceptualizations in the relevant literatures, and the added value of the paradigm

-make a clear distinction between the ‘ideal database’ (ideal from a conceptual perspective), and the actual results

-discuss the conclusions, results and methodology in light of existing findings and the state-of-the-art in the literature

-perform a kind of sensitivity analysis to estimate the impact of spatial aggregation of countries and ports

References

1. Vanoutrive, T. & Zuidhof, P. W. Port Economics in Search of an Audience: The Public Life of Marginal Social Cost Pricing for Ports, 1970-2000. *oecconomia* 9, 371–394 (2019).
2. Raa, T. T. A neoclassical analysis of total factor productivity using input-output prices. in Wassily Leontief and Input-Output Economics (eds. Dietzenbacher, E. & Lahr, M. L.) 151–165 (Cambridge

- University Press, 2004). doi:10.1017/CBO9780511493522.011.
3. Chang, Y.-T., Shin, S.-H. & Lee, P. T.-W. Economic impact of port sectors on South African economy: An input–output analysis. *Transport Policy* 35, 333–340 (2014).
 4. Kwak, S.-J., Yoo, S.-H. & Chang, J.-I. The role of the maritime industry in the Korean national economy: an input–output analysis. *Marine Policy* 29, 371–383 (2005).
 5. Wang, Y. & Wang, N. The role of the port industry in China’s national economy: An input–output analysis. *Transport Policy* 78, 1–7 (2019).
 6. Verschuur, J., Koks, E. E. & Hall, J. W. Port disruptions due to natural disasters: Insights into port and logistics resilience. *Transportation Research Part D: Transport and Environment* 85, 102393 (2020).
 7. Rose, A. & Wei, D. ESTIMATING THE ECONOMIC CONSEQUENCES OF A PORT SHUTDOWN: THE SPECIAL ROLE OF RESILIENCE. *Economic Systems Research* 25, 212–232 (2013).
 8. ics-shipping. Shipping and world trade: driving prosperity. <https://www.ics-shipping.org/shipping-fact/shipping-and-world-trade-driving-prosperity/>.
 9. Statista Research Department. Container shipping - statistics & facts. <https://www.statista.com/topics/1367/container-shipping/> (2021).
 10. World Ocean Observatory. Sea Trade: How Do We Value the Ocean? <https://medium.com/world-ocean-forum/sea-trade-how-do-we-value-the-ocean-aa1c61fff9f7> (2018).
 11. Wiedmann, T., Wilting, H. C., Lenzen, M., Lutter, S. & Palm, V. Quo Vadis MRIO? Methodological, data and institutional requirements for multi-region input–output analysis. *Ecological Economics* 70, 1937–1945 (2011).
 12. Johnson, R. C. & Noguera, G. Accounting for intermediates: Production sharing and trade in value added. *Journal of International Economics* 86, 224–236 (2012).
 13. Fujita, M. & Mori, T. The role of ports in the making of major cities: Self-agglomeration and hub-effect. *Journal of Development Economics* 49, 93–120 (1996).
 14. Thisse, J.-F. How Transport Costs Shape the Spatial Pattern of Economic Activity. (OECD-ITF Joint Transport Research Centre Discussion Paper No. 2009-13., 2009).
 15. Ducruet, C., Notteboom, T. E. & De Langen, P. W. Revisiting Inter-Port Relationships under the New Economic Geography Research Framework. in *Ports in Proximity: Competition and Coordination among Adjacent Seaports* (eds. Notteboom, T. E., Ducruet, C. & De Langen, P. W.) 11–27 (Ashgate, 2009).
 16. Krugman, P. Where in the World is the ‘New Economic Geography’? in *The Oxford Handbook of Economic Geography* (eds. Clark, G. L., Feldman, M. P. & Gertler, M. S.) 49–60 (Oxford University Press, 2000).
 17. Su, B. & Ang, B. W. Input–output analysis of CO2 emissions embodied in trade: The effects of spatial aggregation. *Ecological Economics* 70, 10–18 (2010).
 18. James, R. D. & Campbell, H. S. The effects of space and scale on unconditional beta convergence: test results from the United States, 1970–2004. *GeoJournal* 78, 803–815 (2013).
 19. Taylor, P. J. *World city network: a global urban analysis*. (Routledge, 2004).
 20. Taylor, P. J. Being Economical with the Geography. *Environ Plan A* 33, 949–954 (2001).
 21. Robinson, R. Ports as elements in value-driven chain systems: the new paradigm. *Maritime Policy & Management* 29, 241–255 (2002).
 22. Olivier, D. & Slack, B. Rethinking the Port. *Environ Plan A* 38, 1409–1427 (2006).
 23. Dorsser, C. V., Wolters, M. & Wee, B. V. A Very Long Term Forecast of the Port Throughput in the Le Havre – Hamburg Range up to 2100. *European Journal of Transport and Infrastructure Research* Vol 12 No 1 (2012) (2012) doi:10.18757/EJTIR.2012.12.1.2951.
 24. Notteboom *, T. E. & Rodrigue, J.-P. Port regionalization: towards a new phase in port development. *Maritime Policy & Management* 32, 297–313 (2005).
 25. Goldfarb, D. & Thériault, L. Canada’s ‘Missing’ Trade With Asia. (The Conference Board of Canada, 2008).
 26. UN. An overview of National Compilation and Dissemination Practices Updated Chapter 1 of International Merchandise Trade Statistics: Supplement to the Compilers Manual, United Nations publications, Sales number: 98.XVII.16.,.

Reviewer comments, second round review: –

Reviewer #1:

Remarks to the Author:

Thanks for the revised paper, it is much improved i.m.o.

Some further suggestions

-I am not native in english, i find the text sometimes very dense and hard to digest. For instance (83-84)

'Hence, to date, there is still a technical challenge in combining a top-down representation of the global economy with the bottom-up transportation network used to support the functioning of the global economy' can perhaps be re-written to make it easier to understand.

Even suppl fig 1 is a bit hard to digest (only the 3rd layer; why is the upper right box red [domestic]? That could be any of the four boxes, correct? Why is there a link between final consumption Cd and Industry Cc, but no similar link between Cb and Ca or others? Why is the port termed Cc? Why is there one port in the figure and not 2?

Port handling costs are port-specific, and which we add to individual ports based on a database of port handling costs provided by ITF-OECD14.

-I am not so convinced about the (newly introduced?) term "sealed" (246) where this is defined as low maritime connectivity. It makes more intuitive sense to use it for all islands, as they have no land transport options for trade.

- (294-296): 'The trade unevenness differs considerably per sector (Supplementary Table 2). The largest unevenness is found for the exports of Textiles and Wearing Apparel (sector 5), manufacturing of Transport Equipment (sector 10) and Other Manufacturing (sector 11) with 10%, 50% and 90% of trade flows concentrated in 7 (92), 13 (99) and 15 (124) ports, respectively.' Is there a mistake in this sentence?

-figure 4: I have a hard time understanding why Cork would be red. Is this correct?

On the OxMarTrans model, some issues/questions:

-the port cost data. I could not find this data in the referenced source. The costs differ per type of good, is that reflected in the model?

-The paper mentions: 'In case certain ports are underutilised (allocation lower than port capacity) and others are overutilized (allocated more than port capacity), rebalance the allocation such that the error between the initial allocation and the port capacities is minimized.' How was the capacity of the port determined? This is by no means trivial, as capacity depends on ship size etc. so is not a given fixed number. Furthermore, as far as I am aware, there is no publicly available data on capacity across all ports and segments.

-the data for the EU seems a mistake'For the European Union (EU) countries, the share of maritime transport in extra-EU trade was 53% for imports and 8% for exports in 2015. Our model prediction estimates these shares to be 45.4% for imports and 45.6% for exports in 2015'.

-the cost data reported in Supplementary Table 7 are counter intuitive. The source document is OK, but it may be worthwhile to double check.

-one element which is not included but has traditionally been very relevant in the pricing (i.e. costs) of maritime transport is the imbalance/ empty leg issue. Because of this, waste was exported from Europe and the US to Asia at very cheap rates. In 2015 the rates asia europe were much higher than europe asia. Maybe it is worth mentioning this or making an attempt to incorporate this effect.

Reviewer #2:

Remarks to the Author:

The authors made a real effort to take into account the reviews in their revised manuscript. Among

others the decision to leave out "specialisation/coreness" has resulted in a text that is more focused and sound.

Although much can be said about the claim that "Perhaps we distinguish ourselves here as being opportunistic economists instead of neoclassical or input-output economists. We simply took, to our knowledge, the best available data and methods to find an answer to our research problem.", which downplays the importance of paradigmatic differences and the value-ladenness of concepts and models, the argument that "it is beyond the scope of the paper to talk about the introduction of a new way of thinking in port economics" can be read as a confirmation of the importance of this (i.e. it "would require a full-blown literature review and historical assessment"). As a result, I do not insist to add a broad reflection on this potential contribution of the paper under review. Finally, methodological remarks are taken into account.

details

-391 "on maritime imports to support final demand is increasing. As an complementary metric to"

-> "a complementary metric"

-457-458 "while SIDS have a twice higher import dependency on maritime imports than non-SIDS.": is "import dependency on maritime imports " the same as "maritime import dependency"?

-573-574 "globally that are being allocated on the maritime transport network. A external validation of a model results is included" -> "An external", "of model results"

We would like to thank both reviewers for going through the paper again and providing some additional comments to improve the manuscript. Please see below for a detailed response per comment.

Reviewer #1 (Remarks to the Author):

Thanks for the revised paper, it is much improved i.m.o.

Some further suggestions

-I am not native in english, i find the text sometimes very dense and hard to digest. For instance (83-84)

'Hence, to date, there is still a technical challenge in combining a top-down representation of the global economy with the bottom-up transportation network used to support the functioning of the global economy' can perhaps be re-written to make it easier to understand.

We thank the reviewer for this comment. We have gone through the text again and tried to simplify some of the sentences, including the sentence highlighted by the reviewer:

“Hence, to date, there is still a spatial mismatch between information describing the structure of the global economy (i.e. global trade and supply-chain data) and a bottom up representation of the transportation network used (i.e. observed maritime transport flows) to facilitate this economic structure.”

Even suppl fig 1 is a bit hard to digest (only the 3rd layer; why is the upper right box red [domestic]? That could be any of the four boxes, correct? Why is there a link between final consumption Cd and Industry Cc, but no similar link between Cb and Ca or others? Why is the port termed Cc? Why is there one port in the figure and not 2?

Here, we intended to capture the different components of the link between ports and IO tables by describing all the economic flows which we use throughout the paper/methodology. We focus on one port as the link between ports and the economy/supply-chain is done on a port-by-port basis.

Here we sketched out an example port in country C (Cc) that receives goods from country A and B, which were produced using intermediate production (Ind) in these countries. The flows going through port Cc are then being used in Ind production in country C (which are then further exported or consumed) or go directly into final consumption in country C. Alternatively, it could be that the flows are going to another country, country D, which is dependent on port C via an hinterland transport network.

We can further make an distinction between domestic economic flows (e.g. flows between the port and the country it is located it) or foreign economic flows (flows coming from or going to foreign economies). Similarly, we can distinguish between direct and indirect backward/forward flows. Direct flows feed/come out of the port directly, while economic flows further downstream/upstream in the supply chain are considered indirect flows.

We agree that this figure is unclear and have now added this clarification to the figure legend.

Port handling costs are port-specific, and which we add to individual ports based on a database of port handling costs provided by ITF-OECD14.

This sentence should be:

“Port handling costs vary per good and across ports. We add vessel specific port handling costs to the individual port based on a database provided by the ITF-OECD (not publicly available), which is constructed as part of the ITF global freight model development. “

-I am not so convinced about the (newly introduced?) term “sealocked” (246) where this is defined as low maritime connectivity. It makes more intuitive sense to use it for all islands, as they have no land transport options for trade.

Sealocked is a term introduced by UNCTAD to distinguish between countries that have very low maritime connectivity (e.g. only served by a small number of liner companies). This does not hold for all islands, e.g. some islands in the Caribbean are well connected.

We agree that this term is not commonly used and to avoid jargon we have decided to remove this term throughout.

- (294-296): 'The trade unevenness differs considerably per sector (Supplementary Table 2). The largest unevenness is found for the exports of Textiles and Wearing Apparel (sector 5), manufacturing of Transport Equipment (sector 10) and Other Manufacturing (sector 11) with 10%, 50% and 90% of trade flows concentrated in 7 (92), 13 (99) and 15 (124) ports, respectively.' Is there a mistake in this sentence?

Indeed the sentence was incomplete (same as the one below), as we forgot to mention that this refers to imports and exports.

It should be:

“The largest unevenness is found for the exports of Textiles and Wearing Apparel (sector 5), manufacturing of Transport Equipment (sector 10) and Other Manufacturing (sector 11) with 10%, 50% and 90% of trade flows concentrated in 7 (92), 13 (99) and 15 (124) ports, respectively, for maritime imports (exports).”

However, we simplified this paragraph later on to only mention the sectors and not provide the numbers.

“The trade unevenness differs considerably per sector (see Supplementary Table 2). The largest unevenness is found for the exports of Textiles and Wearing Apparel (sector 5), manufacturing of Transport Equipment (sector 10) and Other Manufacturing (sector 11) while the lowest level of trade unevenness is found for the imports of Agricultural products (sector 1), Food and Beverages (sector 4), and Petroleum, Chemical and Non-Metallic Mineral products. “

-figure 4: I have a hard time understanding why Cork would be red. Is this correct?

The colour red for Cork is indeed the model output, but does require some reflection.

First, the transport model overpredicts the freight flows going through Cork and underpredicts the freight flows through Dublin (flows at Cork and Dublin are similar, while in reality Dublin is larger than Cork). But given the global scale character of the transport model such inconsistencies are to be expected. The reason that, despite, the relatively small trade flow (in weight terms), the trade flow is still quite important is because of the value density of maritime trade going to and from Ireland.

The port of Cork and Dublin have value density of around 2300 USD/tonnes (likely due to the high share of pharmaceutical manufacturing products that go to and from Ireland which have high value density), which is much higher than other ports like Rotterdam (~1000 USD/tonnes) and Singapore (~800 USD/tonnes). This makes both ports, which are not important in weight terms (top 300 ports globally), still important in value terms (top 100 ports in value terms).

Because the input-output tables are also expressed in value terms, the port of Dublin and Cork both make it above the threshold of 0.1% of global industry output being dependent on trade flowing through these port. We do have to acknowledge that the threshold for being important globally (0.1%) is relatively low (~100 ports fall into this category). Apart from that, both ports are essential for domestic industry output >10% and hence the combination of the two makes it a 'red' port, or in other words, a port considered being important for the global and domestic economy. Now, if the throughput through Cork would be exactly the official statistics, it could be that the number would be slightly below the 0.1% as it is now very close to it (0.12%). To avoid confusion, as we admit that it is non-intuitive, we have changed the plot and highlighted another port which is essential for the domestic economy, which was the initial purpose of highlighting Cork. This port is now Victoria in the Seychelles, which is more intuitive.

On the OxMarTrans model, some issues/questions:

-the port cost data. I could not find this data in the referenced source. The costs differ per type of good, is that reflected in the model?

The port cost data, which is mainly the port handling costs, is derived from a database constructed by the ITF-OECD as part of their ITF Freight flow model developed. This database is not open source and we are therefore not at the liberty of sharing this. However, we expect that this dataset can be requested from them relatively easy if desired.

The cost do differ per vessel type, though this is only marginally captured in the database. It is mainly used as a way to scale the port handling costs between ports, making one port more cost-efficient to use than others.

See our sentence above how we have put this down now.

-The paper mentions: 'In case certain ports are underutilised (allocation lower than port capacity) and others are overutilized (allocated more than port capacity), rebalance the allocation such that the error between the initial allocation and the port capacities is minimized.' How was the capacity of the port determined? This is by no means trivial, as capacity depends on ship size etc. so is not a given fixed number. Furthermore, as far as I am aware, there is no publicly available data on capacity across all ports and segments.

This is a very important point indeed and we realised that we forgot to include a description in the methodology.

As the reviewer points out, port capacity is difficult to capture and different types of definitions exist; the maximum instantaneous capacity (theoretical equipment capacity), maximum annual capacity (maximum capacity if berths have 100% occupation) or optimum annual capacity (either economic optimum or operational optimum). We are not aware of any dataset that estimates these capacity per port, nor a commonly agreed best definition for transport modelling applications.

We take a practical approach here and look at the what we call the “theoretical operational capacity” of ports by doing two things. For all vessel types, we look at the capacity weighted weekly arrivals (total capacity called per week) using the 2 years of AIS data. For the 2 year period, we then estimate the utilization rate of the vessel type in a port. We consider that within the 2 year period, the port has reached its theoretical maximum operational capacity for at least a week. We then estimate the trade flows going through the port (in and out) for the 2 year and divide this by 2, ending up with the average annual freight handled. We divide this by the utilization factor to end up with a theoretical operational capacity of the port, which, to us, is a good compromise between something that makes intuitive sense, is consistent across ports, and allows us to make predictions close to predicted trade flows from observed data.

We have now added the following sentences to the methodology:

“

At every port, a capacity is added per vessel type based on available AIS data. Port capacity is non-trivial, as many definitions exist (e.g. maximum instantaneous capacity, maximum annual capacity, optimum annual capacity), and there is no agreed standard definition, nor a global database, on port capacity. Here, we derive a globally consistent estimate of what we call the maximum operational capacity (MOC) of the port; the operational capacity which a port can theoretically reach within existing operational capabilities. First, per vessel type, we estimate the utilization rate (UR) of the port by looking at the weekly port capacity called at ports (multiply the number of calls with the deadweight tonnage of the vessels), and dividing the median with the maximum weekly port capacity called. We then estimate the yearly trade flow (TF) (incoming and outgoing flows) per port and vessel type using the methodology discussed in previous work using AIS data over the period 2019-2020¹⁹. The TF is then divided by the UR rate to derive the MOC per port (p), vessel type (t) and flow direction (f):

$$MOC_{p,t,f} = \frac{TF_{p,t,f}}{UR_{p,t}}$$

“

-the data for the EU seems a mistake 'For the European Union (EU) countries, the share of maritime transport in extra-EU trade was 53% for imports and 8% for exports in 2015. Our model prediction estimates these shares to be 45.4% for imports and 45.6% for exports in 2015'.

Thanks for pointing this out, it should indeed be “53.0% for imports and 48.1% for exports in 2015”

-the cost data reported in Supplementary Table 7 are counter intuitive. The source document is OK, but it may be worthwhile to double check.

We have double checked all values and they are in line with the cited articles/reports.

-one element which is not included but has traditionally been very relevant in the pricing (i.e. costs) of maritime transport is the imbalance/ empty leg issue. Because of this, waste was exported from Europe and the US to Asia at very cheap rates. In 2015 the rates asia europe were much higher than europe asia. Maybe it is worth mentioning this or making an attempt to incorporate this effect.

We are aware that this is indeed an important factor. However, capturing imbalance/empty legs in our model is not feasible as one should go from an aggregated flow model (like ours) to a shipment specific optimization model, which is much more complicated and relies on detailed data. We have mentioned that this is not included in our model in the methodology now:

“The cost elements included capture the different components that together determine freight costs. However, we still ignore important factors that could not be included because of data limitations, such as costs associated with imbalances/empty legs or other route varying freight cost components, which lead to high freight costs in some regions (e.g. Pacific islands).”

Reviewer #2 (Remarks to the Author):

The authors made a real effort to take into account the reviews in their revised manuscript. Among others the decision to leave out "specialisation/coreness" has resulted in a text that is more focused and sound.

Although much can be said about the claim that "Perhaps we distinguish ourselves here as being opportunistic economists instead of neoclassical or input-output economists. We simply took, to our knowledge, the best available data and methods to find an answer to our research problem.", which downplays the importance of paradigmatic differences and the value-ladenness of concepts and models, the argument that "it is beyond the scope of the paper to talk about the introduction of a new way of thinking in port economics" can be read as a confirmation of the importance of this (i.e. it "would require a full-blown literature review and historical assessment"). As a result, I do not insist to add a broad reflection on this potential contribution of the paper under review.

Finally, methodological remarks are taken into account.

details

-391 "on maritime imports to support final demand is increasing. As an complementary metric to" -> "a complementary metric"

Thank you, this is changed now.

-457-458 "while SIDS have a twice higher import dependency on maritime imports than non-SIDS.": is "import dependency on maritime imports " the same as "maritime import dependency"?

Yes that is the same, we have changed this now.

-573-574 "globally that are being allocated on the maritime transport network. A external validation of a model results is included" -> "An external", "of model results"

Thank you, this is changed now.